# Polydatin: Pharmacological Mechanisms, Therapeutic Targets, Biological Activities, and Health Benefits

**DOI:** 10.3390/molecules27196474

**Published:** 2022-10-01

**Authors:** Ahmad Karami, Sajad Fakhri, Leila Kooshki, Haroon Khan

**Affiliations:** 1Student Research Committee, Faculty of Pharmacy, Kermanshah University of Medical Sciences, Kermanshah 6714415153, Iran; 2Pharmaceutical Sciences Research Center, Health Institute, Kermanshah University of Medical Sciences, Kermanshah 6734667149, Iran; 3Department of Pharmacy, Abdul Wali Khan University Mardan, Mardan 23200, Pakistan

**Keywords:** polydatin, pharmacological mechanism, therapeutic target, health benefits, novel delivery system

## Abstract

Polydatin is a natural potent stilbenoid polyphenol and a resveratrol derivative with improved bioavailability. Polydatin possesses potential biological activities predominantly through the modulation of pivotal signaling pathways involved in inflammation, oxidative stress, and apoptosis. Various imperative biological activities have been suggested for polydatin towards promising therapeutic effects, including anticancer, cardioprotective, anti-diabetic, gastroprotective, hepatoprotective, neuroprotective, anti-microbial, as well as health-promoting roles on the renal system, the respiratory system, rheumatoid diseases, the skeletal system, and women’s health. In the present study, the therapeutic targets, biological activities, pharmacological mechanisms, and health benefits of polydatin are reviewed to provide new insights to researchers. The need to develop further clinical trials and novel delivery systems of polydatin is also considered to reveal new insights to researchers.

## 1. Introduction

Polydatin is a stilbenoid polyphenol and a monocrystalline natural compound [1,2,3]. The plant families *Vitaceae*, *Liliaceae*, and *Leguminosae* are prominent sources of polydatin extraction [4]. It is primarily isolated from the rhizome and root of *Polygonum cuspidatum* [5], traditionally used against inflammation, infection, jaundice, skin burns, and hyperlipemia [6]. *Reynoutria japonica* is an invasive plant from the Far East, and a main source of polydatin in industrial scale [7]. This plant source of polydatin is so invasive that it has reached even the most remote mountain areas where it unbalances the ecosystem including the agricultural potential of mountain areas, being advantaged by the context of climate change [8]. This phytochemical is also found in other plant sources such as red wine [9], nuts, vegetables, fruits [5], hop cones/pellets, and cocoa- and chocolate-containing products [10].

Polydatin is a glucoside derivative of resveratrol. Moreover, polydatin has a higher antioxidant [3] and anti-inflammatory [11] activity compared to resveratrol [1,12,13,14,15,16]. It possesses four leading derivatives in nature, including *trans*-polydatin, *trans*-resveratrol, *cis*-polydatin, and *cis*-resveratrol [10,17]. *trans*-polydatin, itself, can be produced from 4-coumaryol-CoA through a stilbene synthase reaction (Figure 1). Several methods have been developed for the isolation of resveratrol from other isomers in *P. cuspidatum*, including reflux extraction, filtering, hydrolyzing, liquid–liquid extraction, eluting, and high-performance liquid chromatography coupled to an ultraviolet−visible diode array detector [18,19].

The therapeutic and protective effects of polydatin mainly originate from its anti-inflammatory, antioxidant, and anti-apoptotic activities [20]. Previous studies have shown a wide range of therapeutic effects of polydatin in the treatment of several pathological diseases such as cancer [21], cardiovascular diseases [22], diabetes [23], neurodegenerative diseases [24], hepatic/respiratory diseases [25,26], gastrointestinal diseases [27], infectious diseases [28], rheumatoid diseases [29], and skeletal/women disorders [30,31].

Previously, the protective effects of polydatin have been disclosed against neurodegenerative diseases [20,32,33], atherosclerosis [34,35], and multiple organ ischemia/reperfusion injury [36]. The protective effects of polydatin on damaged macrophages were also highlighted by Liu et al. [37]. A more recent study also reviewed the pharmacological effects of polydatin in some diseases [38]. In the current mechanistic review, the therapeutic targets, pharmacological mechanisms, biological activities, and health benefits of polydatin are highlighted for all related biological/pathological conditions. Moreover, the need to provide different novel delivery systems and clinical trials of polydatin is also considered to reveal new insights to researchers.

## 2. Pharmacological Mechanisms of Polydatin

Considering the critical role of inflammation, oxidative stress, and apoptosis in the progression of several diseases, polydatin could play therapeutic roles by regulating associated signaling pathways.

### 2.1. Anti-Inflammatory Effects

As a protective mechanism, the initiation of inflammation is attributed to the involvement of immune system response against diverse factors such as pathogens, injured tissues, and infectious agents, which is generally attributed to favoring tissue repair and organ healing. This defense process is divided into two major categories known as acute and chronic inflammation. At present, chronic inflammation could be considered a vital cause of human diseases (e.g., atherosclerosis, arthritis, diabetes, neurodegenerative diseases, and cancer) [39,40]. In this context, chronic inflammation is induced via uncontrolled acute inflammatory responses through inflammatory mediators such as cytokine production and immune cell recruitment [39,41]. Reportedly, nuclear factor-kappa B (NF-κB) is considered as the key transcription factor in increasing inflammatory mediators/cascades, including tumor necrosis factor-α (TNF-α), interleukin-1 beta (IL-1β), IL-6, and intercellular adhesion molecules (ICAMs) in various diseases. Such inflammatory mediators and interconnected pathways are mainly involved in the underlying mechanisms of the inflammatory state [42]. Several studies show that phytochemicals are multi-target agents that effectively represent their anti-inflammatory activities through underlying cellular mechanisms against different diseases associated with chronic inflammation [43]. In the phytochemical context, it has been reported that polydatin is a potent anti-inflammatory plant secondary metabolite [44], beneficially promoting miR-200a expression to regulate the Kelch-like ECH-associated protein 1 (Keap1)/nuclear factor E2-related factor 2 (Nrf2) antioxidant axis. This pathway, in turn, suppresses nucleotide-binding domain-like receptor protein 3 (NLRP3) inflammasome activation against diverse chronic inflammation-related diseases in vivo [25,45]. Furthermore, under inflammatory conditions, polydatin exhibited anti-inflammatory roles by improving AMP-activated protein kinase (AMPK)/sirtuin1 (Sirt1)/Nrf2 signaling expression, leading to the inhibition of the NF-κB/inhibitor of nuclear factor-kappa B α (IκBα)/NLRP3 pathway, as well as the reduction of pro-inflammatory cytokines such as TNF-α, IL-1β, and IL-6 [31,46,47]. Meanwhile, polydatin treatment downregulated the expression of cerebral ischemia or spinal cord injury (SCI)-induced inflammatory mediators such as Toll-like receptor 4 (TLR4), NF-κB, cyclooxygenase-2 (COX-2), inducible nitric oxide synthase (iNOS), nitric oxide (NO), and ICAM-1 [48,49,50]. In line with this, polydatin potentially plays a vital role in preventing the inflammatory action by reducing mitogen-activated protein kinases (MAPKs). This response is mainly orchestrated by extracellular-signal-regulated kinases (ERK1/2), c-Jun N-terminal kinase 1/2 (JNK1/2), and p38 protein kinases, and consequently inhibits NF-κB p65 phosphorylation and the release of inflammatory factors such as xanthine oxidase (XOD), prostaglandin E2 (PGE2), TNF-α, IL-1β, and COX-2 [51,52]. Another mechanism of the neuroprotective effect of polydatin is mediated by the CCAAT/enhancer-binding proteinβ (C/EBPβ)/metastasis-associated lung adenocarcinoma transcript 1 (MALAT1)/cAMP response element binding (CREB)/peroxisome proliferator-activated receptor gamma co-activator 1α (PGC-1α)/peroxisome proliferative-activated receptor γ (PPARγ) signaling pathway. This process results in silencing NF-κB-associated downstream inflammatory mediators, which could alleviate cerebral infarct volume and ameliorate the integrity of the blood–brain barrier (BBB) [53]. Numerous studies have confirmed that polydatin application could suppress phospholipase A2 (PLA2) against lipopolysaccharide (LPS)-induced lung injury [54], decreasing the serum levels of alanine transaminase (ALT) and aspartate aminotransferase (AST) against fulminant hepatic failure (FHF) [55]. Additional reports also revealed that polydatin downregulated the retinoic acid receptor-related orphan receptor gamma t (RORγt) and signal transducer and activator of transcription 3 (STAT3) gene expression, attenuated IL-17 production against CD3/DC28-induced peripheral blood mononuclear cells and arthritis [11,56], and ultimately reduced both the NF-κB and vascular endothelial growth factor (VEGF) as well as the circulating levels of the downstream inflammatory cascade including IL-6, TNF-α [56].

Altogether, polydatin modulates several inflammatory mediators to show potential anti-inflammatory effects in various diseases.

### 2.2. Antioxidant Effects

Oxidative stress is characterized by aberrant generation of the reactive oxygen species (ROS) and reactive nitrogen species (RNS) as byproducts of this metabolic process which leads to devastating effects on molecular intracellular signaling pathways and to the development of numerous diseases [57,58]. The antioxidant reaction of polydatin is presented via reducing lipid peroxidation while promoting antioxidant enzymatic activity against hepatotoxicity induction in vivo [59]. Polydatin has also shown modulatory roles on ROS and RNS in peripheral [60] and central diseases [49]. Nrf2 could be the most vital antioxidative response-associated transcription factor and alleviates inflammation [61]. In this regard, polydatin strongly acts as an antioxidant agent on LPS-induced BV2 microglia cells in SCI-induced rat models via promoting the Nrf2/heme oxygenase-1 (HO-1) pathway activity, resulting in improved locomotor performance, suppressing spinal edema, and attenuating neurological deficits [24]. Additionally, the activation of protein kinase B (Akt) phosphorylation may be accelerated through the Nrf2/HO-1/NAD(P)H quinone dehydrogenase 1 (NQO1) expression-mediated antioxidant pathway after polydatin administration [51]. It has also been proposed that polydatin could significantly enhance Nrf2/thioredoxin (TRX) antioxidant signaling, and upregulate the Gli1/patched 1 (Ptch1)/superoxide dismutase 1 (SOD1) pathway, indicating its neuro/nephron-protective action against ischemic brain/renal injury-induced ROS generation [62,63,64]. Further in vivo studies have also indicated that polydatin supplementation provided an antioxidant ability through the Sirt1/Nrf2/antioxidant-responsive element (ARE) signaling pathway to reduce diabetes-induced renal dysfunction [65], as well as through the Notch1/Hes1-Pten/Akt axis to improve ischemic/reperfusion-injured diabetic heart disease [66,67]. Further, polydatin consumption is possibility beneficial against atherosclerosis and cardiovascular disease-induced oxidative stress via the scavenging of hydroxyl, oxygen free radicals, myeloperoxidase (MPO), and ROS [56,68,69], elevating enzymatic antioxidants such as SOD, glutathione peroxidase (GSH-Px), glutathione transferase (GST), catalase (CAT), and glutathione (GSH) [70]. Polydatin also induced protein kinase C (PKC) and the mitochondrial adenosine triphosphate (ATP)-sensitive-K^+^ (mito-K_ATP_) channel [71] towards the modulation of mitochondrial function and the Akt signaling pathway [35,72]. The body of this evidence strongly supports that the attenuation of Akt is associated to the antioxidant effect of polydatin [50]. In addition, polydatin attenuated the activity of hepatic stellate cells (HSCs) through its antioxidant activity on sphingosine kinase 1 (SphK1) signaling to ameliorate mice liver fibrosis induced by carbon tetrachloride (CCl4) [73].

Overall, polydatin employs several antioxidant mediators while suppressing oxidative pathways towards therapeutic responses in several disorders.

### 2.3. Anti-Apoptotic Effects

Apoptosis is defined as a programmed model of physiological cell death without the involvement of an inflammatory condition. Nonetheless, the uncontrolled regulation of this mechanism could be engaged in several diseases, including cancer, autoimmune diseases, and neurodegenerative disorders [74]. In this regard, polydatin significantly exhibited anti-apoptotic effects in a mice model of liver injury through the attenuation of Bcl-2-associated x (Bax) expression and enhancing B-cell lymphoma 2 (Bcl-2) expression [59]. In another experimental study, polydatin showed its anti-apoptotic activity by increasing Bcl-2 or D-cyclins and reducing caspase-3 or Bax levels in gastrointestinal injury. Such an effect is made by the activation of Gli1 transcription factor followed by the upregulation of the Sonic hedgehog (Shh) signaling pathway as the major component in repairing dextran sulfate sodium-induced colitis-related damaged tissues [75]. In another study, polydatin exerted its protective effects against acute lung injury-induced mitochondrial apoptosis by increasing Parkin-mediated mitophagy expression via inhibiting Bax, cytochrome c, and caspase-3 activity as well as promoting Bcl-2 and mitochondrial membrane potential proteins [76]. Furthermore, polydatin administration could attenuate neuronal apoptosis through repressing p53/MAPK/JNK signaling activation in a rat model of ischemic brain injury [62]. Consequently, polydatin improved autophagy and apoptosis during osteoarthritis through suppressing the MAPK and phosphoinositide 3-kinases (PI3K)/Akt/mammalian target of rapamycin (mTOR) signaling pathway [77]. Taken together, the described anti-apoptotic activities of polydatin indicate that this compound could be considered as a promising agent for apoptosis-induced tissue damage.

## 3. Biological Activities of Polydatin

As previously mentioned, polydatin exhibits significant pharmacological effects through its anti-inflammatory, antioxidant, and apoptosis-modifying activities. During different physiological/pathological conditions, the activity and expression of involved signaling pathways would change and polydatin plays a critical modulatory effect. We previously described near interconnections between the aforementioned signaling pathways during diseases which support each other [78,79]. Such effects make polydatin a potential therapeutic compound in the treatment of different pathological conditions.

### 3.1. Anticancer Effects

According to statistics from the World Health Organization, cancers leave the heaviest strain on the worldwide population based on the disability-adjusted life year (DALY) scale [80]. Radiotherapy, surgery, and chemotherapy are the main strategies in treating various cancer types [81,82]; however, associated resistance and side effects limit their applications. In addition, considering the multiple signaling pathways involved in cancer pathogenesis, there is an urgent need to provide novel alternatives and safe multi-target phytochemicals [83,84,85]. In this regard, the anticancer activity of polydatin has been elucidated against different cancer types [86].

From the mechanistic point, polydatin modulates oxidative stress to decrease carcinogenesis and mutagenesis [87]. It is reported that polydatin inhibits tumor growth [88] and improves radiosensitivity via inducing apoptosis in colorectal cancer models [89]. Polydatin also plays therapeutic roles in the treatment of leukemia alone or in combination with Janus kinase (JAK) inhibitors [86,90]. Moreover, it is revealed that polydatin is efficacious in the treatment of laryngeal cancer [91], nasopharyngeal cancer [92], ovarian cancer [93], lung cancer [94,95,96], glioblastoma multiforme [97], multiple myeloma [98], and cervical cancer [99,100]. Consequently, polydatin exerts therapeutic effects against such cancers employing different mechanisms, including the inhibition of proliferation/migration [97]. Several pre-clinical studies demonstrated that polydatin exerted therapeutic effects against an orthotopic metastatic tongue cancer model via inhibiting glucose 6 phosphate dehydrogenase (G6PD), and reduced tumor size and lymph node size and metastases [1]. In another study, the inhibition of G6PD by polydatin reduced tyrosine kinase inhibitor (TKI) resistance in breast cancer models [21]. In addition, polydatin suppressed the growth of MDA-MB-231 and MCF-7 breast cancer cell lines in vitro [101]. This secondary metabolite also induced apoptosis and inhibited cell proliferation/migration in breast cancer cell lines when used in combination with 2-deoxy-D-glucose [102].

Osteosarcoma is a malignant bone tumor; however, its etiology and pathogenesis remain unclear [103]. Prevailing in vitro studies showed that polydatin inhibited cells’ migration [103] and proliferation [103,104,105]. In this regard, inhibition of the β-catenin signaling pathway by polydatin is associated with its anti-proliferative effects [105]. Polydatin also induced apoptosis [103,104] via different mechanisms such as reducing the expression/phosphorylation of STAT3, increasing autophagy-related gene expression [104], and caspase-3 activity [105] in vitro. Polydatin also exerted therapeutic effects against doxorubicin-resistant osteosarcoma by suppressing TUG1/Akt signaling, promoting apoptosis and suppressing cell proliferation. In addition, polydatin reduced tumor size in animal models [106]. On another point, chronic liver disease and cirrhosis lead to hepatocellular carcinoma (HCC). HCC is a common malignancy of the liver and is classified as the third leading cause of cancer-related deaths in the world [107]. Similarly, polydatin showed positive effects against HCC through the inhibition of the Akt/STAT3–forkhead box protein O1 (FOXO1) signaling pathway towards apoptosis induction [108]. It also inhibited proliferation/invasion/migration of cell lines [107], and reduced tumor growth by increasing caspase-3 expression and terminal deoxynucleotidyl transferase (TdT) dUTP nick-end labeling (TUNEL) activity in animal models of HHC [107].

On the other hand, it is worth mentioning that polydatin could remarkably induce apoptosis against acute monocytic leukemia cell proliferation via modulating cyclin D1, Bcl-2, cyclin A, and Bax, accompanied by cell cycle arrest at the S phase [90]. Additionally, polydatin induced apoptosis in human nasopharyngeal carcinoma CNE cells through ROS-mediated mitochondrial dysfunction and endoplasmic reticulum stress. Thus, polydatin might be a promising anti-tumor agent through inducing apoptosis [92]. Additionally, polydatin inhibited the progression of doxorubicin-resistant osteosarcoma by regulating the taurine-upregulated gene 1 (TUG1)/Akt signaling pathway [106]. Furthermore, suppressing the phosphorylation of CREB and cyclin D1 gene expression might be associated with the apoptotic effect of polydatin on human breast cancer cells [101]. Moreover, extensive studies have demonstrated that the suppression of platelet-derived growth factor (PDGF)/Akt signaling [91], stimulation of cytochrome c release, and activation of poly (ADP-ribose) polymerase (PARP) fragmentation [109] could induce cell cycle arrest and apoptosis on polydatin-treated laryngeal/cervical cancer and oral squamous cell carcinoma.

Not only is polydatin efficacious in the treatment of various cancer types, it also reduces adverse effects of some chemotherapeutic agents such as cisplatin [110] and doxorubicin [111]. Polydatin plays a protective role against cisplatin-induced toxicity via improving antioxidant mechanisms and tissue regeneration [110]. It is reported that polydatin attenuated the cardiotoxicity of doxorubicin by increasing heart rate, arterial pressure, and GSH-Px activity while reducing myocardial injury. In addition, polydatin decreased ST, QT, and QRS intervals in an electrocardiogram. Antioxidant effects and the enhancement of metabolism were also associated with the protective effects of polydatin [111].

Taken together, polydatin could be a potential agent in the treatment of different cancer types. Apoptotic and anti-proliferative effects of polydatin, especially, are associated with its anticancer effects. Considering the protective effects of polydatin against the adverse effects of chemotherapy agents, polydatin might be an excellent choice to be co-administered with other agents. In Table 1, several in vitro and in vivo studies on the anticancer properties of polydatin are summarized.

### 3.2. Cardioprotective Effects

It is reported that different cardiovascular diseases are associated with stress oxidative and nitrosative processes, such as atherosclerosis and irreversible injury after ischemic reperfusion [112]. In addition, the interconnections between inflammatory processes and the symptoms of cardiovascular diseases have been elucidated [113]. As previously described, polydatin demonstrated potential antioxidant and anti-inflammatory effects. Polydatin has been reported to exert therapeutic effects against cardiovascular diseases such as acute myocardial infarction (AMI) [114], atherosclerosis [115], pulmonary hypertension [116], vascular damages [117], and sepsis-induced cardiac injury [118].

Polydatin consistently inhibited ventricular remodeling through neurohormones, especially in the renin–angiotensin system (RAS) and peroxidation [119,120,121]. In animal models of cardiac remodeling, including mice and rats, polydatin reduced cardiac weight in both animals, and decreased angiotensin II (Ang II) in mice. Polydatin also decreased cardiomyocyte size, aldosterone levels, TNF-α, endothelin-1, Ang II, blood pressure, and ventricular collagen volume [119]. Additionally, polydatin also increased NO, improved CAT and GSH-Px activity, and decreased serum hydroxyproline levels [120].

Insulin deficiency in diabetes causes cardiomyopathy, a disorder of the heart’s ventricles. Polydatin reduced ventricular dysfunction, improved mitochondrial bioenergetics, and increased autophagy flux by upregulating Sirt3 [122]. In addition, the inhibition of NAPDH oxidase and NF-κB is attributed to the cardioprotective effects of polydatin against diabetic cardiomyopathy [35]. Diabetes mellitus during myocardial infarction/reperfusion (MI/R) can increase the severity of oxidative/nitrative injury. Polydatin reduced MI/R cardiac damage during diabetes by activating the Notch1/Hes1-dependent Pten/Akt signaling pathway [36,66,123]. Assessment of the effects of polydatin on diabetes-associated myocardial hypertrophy showed a decrease in fasting blood glucose, insulin, HbA1C, weight gain, and insulin resistance levels by inhibiting NF-κB, COX-2, and the iNOS signaling pathway and activating PPARβ. It also improved hypertrophy factors in animal models of myocardial hypertrophy [124].

Studies have shown that polydatin decreased serum levels of total cholesterol (TC), low-density lipoprotein cholesterol (LDL-C), triglycerides (TG), hepatic TG concentrations, and LDL-C/high-density lipoprotein cholesterol (HDL-C) and TC/HDL-C ratios. These effects may indicate the protective effect of polydatin on the liver, coronary heart disease, and atherosclerosis [125,126,127]. Oxidative stress caused by the proliferation of vascular smooth muscle cells (VSMCs) leads to plaque formation and the progression of atherosclerosis. Polydatin inhibited oxidative stress by activating the NOS/Sirt1 pathway [115]. Polydatin also exerted potential protective effects against atherosclerosis by regulating pre-B-cell colony-enhancing factor (PBEF) and the inhibition of cholesterol accumulation in PBEF-induced macrophages. During an in vivo study, polydatin reduced blood lipids and atherosclerotic lesions, regulated the expression of cytokines and genes involved in cholesterol metabolism, and decreased PBEF mRNA and PBEF protein [128]. Moreover, polydatin exerted protective and therapeutic effects against atherosclerosis by inhibiting the PI3K/Akt/mTOR signaling pathway and improving autophagy disorders [129]. Polydatin’s inhibitory effects on β-hydroxy-β-methyl-glutaryl-CoA reductase (HMG-R) and the expression of proprotein convertase subtilisin/kexin type-9 (PCSK-9) are contributed to its anti-atherosclerotic effects [35]. Taken together, polydatin is reported to exert beneficial effects against atherosclerosis mainly through the protection of VSMCs, endothelial cells, mitochondria, and regulatory effects on lipid metabolism and antioxidant and anti-inflammatory effects [34].

In high-fat-diet (HFD)-induced obese mice, polydatin consumption caused weight loss, decreased TC, TG, and LDL, and increased HDL. Polydatin also reduced the size of adipose tissues and the volume of retroperitoneal fat. Polydatin exerted such effects by downregulating PPARγ, upregulating mRNA and leptin expression, and reducing TNF-α/monocyte chemoattractant protein-1 (MCP-1) [130]. Polydatin also inhibited the formation of foam cells caused by peritoneal macrophages via regulating its intracellular metabolism and antioxidant properties by regulating the PPARγ signaling pathway [131].

Polydatin exerted protective effects against cardiac ischemia/reperfusion (I/R) injury through various mechanisms, including the activation of the PKC-K_ATP_-dependent signaling pathway and a direct antioxidative stress effect [71]. Studies have shown that intravenous administration of polydatin before I/R injury limits the size of the infarcted area, creatine phosphokinase, leakage of lactate dehydrogenase from damaged myocardial cells, changes in SOD activity, and malondialdehyde (MDA) levels [132]. Other cardioprotective mechanisms of polydatin include increased Bcl-2 protein expression and decreased Bax protein expression, which inhibits I/R-induced apoptosis in rat myocardium. Additionally, polydatin inhibits the rho kinase (ROCK) system, reducing apoptosis and the size of the infarcted area in animal models [22]. Constitutive nitric oxide synthase (cNOS) and the resulting increase in NO production are other pathways that promote the protective effects of polydatin against I/R injury. The study by Zhang et al. showed that polydatin reduced arrhythmia score and size of myocardial infarction, improved left developed ventricular pressure and coronary flow, increased SOD activity, and decreased MDA compared to the control group [133]. In addition, polydatin limited myocardial damage induced by I/R through increasing and improving autophagic flux [2,134]. Thus, polydatin cleared damaged mitochondria and reduced ROS in cell lines. The results of an in vivo study on cardiac I/R injury showed that polydatin reduced the size of the infarcted area, increased the ejection fraction, and caused higher left fractional ventricular shortening [134].

In the final analysis, polydatin could be considered a potential agent in the treatment of various cardiovascular diseases. Decreasing ROS level, NF-κB activity, VSMC proliferation, and RAS system activity and improving Sirt3 activity and lipid metabolisms contribute to polydatin’s therapeutic and protective cardiovascular effects. The details on the cardioprotective activity of polydatin are presented in Table 2.

### 3.3. Anti-Diabetic Effects

Pancreatic beta cells are sensitive to oxidative stress and, thus, the development of diabetes. Polydatin has been reported to inhibit inflammation and increase the expression of insulin receptor substrate 2 towards insulin sensitivity, insulin secretion, and the improvement of glucose resistance/fat metabolism. Polydatin exhibits such effects through multiple mechanisms, including ROS scavenging, improving functional cell markers, and modulating anti-apoptotic/antioxidant mediators [136,137].

Diabetic nephropathy is a major cause of end-stage renal disease and one of the most important complications of diabetes progression. Polydatin reduces the expression of fibronectin (FN) protein, an important fibrotic factor, in glomerular mesangial cells (GMCs) by inhibiting the NF-κB signaling pathway. Polydatin also reduced the expression of ICAM-1 and TGF-β in glomerular mesangial cells and, thus, exerted its anti-inflammatory effects [138]. Activation of the SphK1/sphingosine 1-phosphate (S1P) signaling pathway by high blood sugar is known as another factor that leads to an increase in the expression of FN protein in glomerular mesangial cells. Accordingly, polydatin prevented an increase in FN and the expression of ICAM-1 under the diabetes condition [139]. In addition, polydatin reduced oxidative stress in glomerular mesangial cells by upregulating the casein kinase 2 interacting protein-1 (CKIP-1)/Nrf2/ARE signaling pathway [23].

Impaired fat metabolism is another problem observed along with impaired glucose metabolism in diabetic patients. Polydatin improved fat/glucose metabolism by regulating PCSK-9 [140] and the Akt signaling pathway towards decreasing fasting blood glucose, glycosylated hemoglobin, glycosylated serum proteins, TC, TG, and LDL, while increasing serum insulin levels in diabetic rats [141].

Cardiovascular problems (e.g., macrovascular and microvascular diseases) are other complications following diabetes [142]. In this regard, polydatin improved endothelial function [72,142,143] in hyperglycemic conditions by activating the PPARβ/NO signaling pathway. Polydatin also improved endothelial damage in the aorta and vasodilation [142]. Diabetic neuropathy is another problem that affects half of diabetes patients. Mitochondrial dysfunction is closely related to the incidence of diabetic neuropathy. Polydatin improved mitochondrial function through the Sirt3/Nrf2 axis and, thus, reduced thermal and mechanical hyperalgesia in animal models [144].

In conclusion, polydatin could be an effective natural compound in the treatment of diabetes and complications related to diabetes. Activation of anti-fibrotic mechanisms and Nrf2-related signaling pathways and inhibition of NF-κB could improve glucose and lipid metabolism following polydatin administration.

### 3.4. Effects on Gastrointestinal Tract

Ulcerative colitis and inflammatory bowel disease are caused/characterized by dysfunction in the intestinal epithelial barrier. In addition to the inflammatory mechanisms involved in this disease, oxidative stress and reactive oxygen and nitrogen species could be involved in tissue damage and intestinal fibrosis [47]. Studies have shown that polydatin has an inhibitory effect on the NF-κB signaling pathway towards therapeutic effects against colitis [47,145,146]. In this regard, polydatin has shown therapeutic effects against ulcerative colitis in animal models by reducing NF-κB expression and increasing Sirt1/Nrf2 expression [47]. Polygonum cuspidatum root extract and its main constituent polydatin showed therapeutic responses against ulcerative colitis by inhibiting the NF-κB signaling pathway [145]. Polydatin also suppressed colitis in animal models by regulating the Shh signaling pathway [75].

Severe hemorrhagic shock causes damage to the small intestine. Increasing free radicals and mitochondrial damage caused by gastrointestinal I/R are among the factors influencing small bowel injury in severe hemorrhagic shock. Polydatin acts on the Sirt1/PGC-1α/SOD2 axis in the small intestine, thereby reducing oxidative stress and small intestinal damage [27,147,148].

Taken together, polydatin is an efficacious compound in the treatment of ulcerative colitis and severe hemorrhagic shock through antioxidant and anti-inflammatory effects. Inhibition of NF-κB is a major mechanism of polydatin towards its gastroprotective effects.

### 3.5. Anti-Microbial Effects

Although polydatin has not shown promising antibacterial effects, it has been shown that polydatin can exert positive antioxidant and anti-inflammatory effects against *Staphylococcus aureus*-induced mastitis and related lipoteichoic acid (LTA)-induced injury. The inhibitory effect of polydatin is that of TLR2-mediated activation of the p38MAPK/NF-κB signaling pathway and production of ROS [28,149]. Similarly, polydatin inhibited the NF-κB signaling pathway, contributing to the suppression of inflammatory responses and the development of pulmonary fibrosis induced by *Mycoplasma pneumoniae* infection. Inhibition of NLRP3 inflammasome by polydatin is the other mechanism that is employed by polydatin against *Mycoplasma pneumoniae* infection [45,150]. It also has been shown that polydatin exerted anti-inflammatory/protective responses against septic lung injury through the upregulation of HO-1 in lung tissue [151]. Since the coronavirus disease of 2019 (COVID-19) pandemic started, the effects of different natural compounds have been studied. It has been revealed that polydatin has shown inhibitory effects on the entry of severe acute respiratory syndrome coronavirus 2 (SARS-CoV-2) into cells. This originated from binding directly to main protease (Mpro) of SARS-CoV-2. In addition, activation of sirtuin by polydatin may contribute to potential anti-COVID-19 effects [152].

To conclude, polydatin might be a good candidate in the treatment of respiratory diseases such as COVID-19 and *Mycoplasma pneumonia*-induced pulmonary fibrosis. Antioxidant and anti-inflammatory effects are primarily related to therapeutic anti-microbial effects of polydatin.

### 3.6. Hepatoprotective Effects

Recent studies have demonstrated polydatin’s hepatoprotective effects against alcoholic liver disease [153], non-alcoholic fatty liver disease (NAFLD) [154], and liver fibrosis [70]. Moreover, polydatin showed beneficial effects in drug-induced liver injury, including acetaminophen-induced injury, through antioxidant and anti-inflammatory effects. In one particular study, polydatin decreased the expression and levels of Bax, cytochrome c, cleaved caspase-3, cleaved caspase-9, and apoptotic protease-activating factor 1, while increasing Bcl-2. Polydatin also reduced oxidative stress factors such as ROS, NO, and MDA [59].

Alcoholic liver disease is one of the problems caused by excessive alcohol consumption, which can occur as a mild steatohepatitis or even severe cirrhosis of the liver [153]. Evaluation of the effect of polydatin on ethanol-induced liver damage in rats showed that polydatin exerted antioxidant and anti-inflammatory effects against ethanol-induced liver damage through downregulation of cytochrome P450 2E1 (CYP2E1), upregulation of Nrf2/HO-1, and modulation of TLR4/NF-κB p65 [48,155].

The mechanisms involved in the most common liver disease, NAFLD, have not been fully identified. Similar to alcoholic liver disease, it can also occur in the form of steatohepatitis in mild cases or in the form of liver cirrhosis in severe cases [156]. Polydatin has shown protective and therapeutic effects against NAFLD via various mechanisms [154]. Further, it reduced animal models’ TG, TC, and free fatty acid (FFA). Polydatin also exerted protective and therapeutic effects against NAFLD by different mechanisms, including lipid peroxidation suppression, reduction of TNF-α, and lipogenesis induced by sterol regulatory element-binding protein (SREBP-1c) [156]. In addition, polydatin decreased serum levels of AST, ALT, caspase-3, TUNEL-positive cells, TG, and oxidative stress in mice. Polydatin downregulated NADPH oxidase 4 (NOX4), inhibited the TLR4/NF-κB p65 signaling pathway, and exerted antioxidant and anti-inflammatory effects in vivo [157].

It has been shown that polydatin can exhibit hepatoprotective effects against carbon tetrachloride and sulfur mustard through antioxidant and anti-inflammatory mechanisms [158,159]. Additional studies elucidated that polydatin has protective and therapeutic effects against liver damage caused by carbon tetrachloride [70,159,160] through reducing liver fibrosis, suppressing inflammatory factors, and improving liver function/histological parameters. It also suppresses hepatic 4-HNE production andNOX4 expression and increases GSH levels, GST activity, SOD activity, GSH superoxidase, and CAT activity [70,159].

FHF is known as a clinical syndrome with high mortality. Consumption of polydatin prior to FHF induction showed protective effects by reducing TNF-α production and inhibiting NF-κB activity. It was revealed that polydatin reduced ALT, AST, mortality, and improved histopathological lesions [55,161].

Taken together, improvement of antioxidant capacity and anti-inflammatory effects are the major reasons of polydatin’s therapeutic effects against liver diseases. These effects are mainly applied through the activation of Nrf2-related signaling and inhibition of NF-κB. Therefore, polydatin is a potential compound in the treatment of liver diseases such as FHF, NAFLD, and alcoholic liver disease. Details on the hepatoprotective activity of polydatin are presented in Table 3.

### 3.7. Neuroprotective Effects

Neurological diseases are known as the second leading cause of death globally. Such disorders are the first cause of severe long-term disability worldwide [162]. Oxidative stress and inflammatory and apoptotic processes play essential roles in the pathogenesis of neurological diseases [112]. Regarding the previously described antioxidant, anti-inflammatory, and anti-apoptotic activities of polydatin, it can be a potential agent in the treatment of neurological diseases. Studies revealed therapeutic and protective effects of polydatin against Parkinson’s disease (PD) [163], intracranial hemorrhage (ICH) [164], cerebral ischemia [5], hemorrhagic shock [165,166], SCI [24], dementia [167], and traumatic brain injury [168].

Oxidative stress and mitochondrial dysfunction are two critical factors involved in PD pathogenesis [163]. It is shown that polydatin has neuroprotective effects against PD through various mechanisms [9,163,169,170]. Polydatin attenuated motor dysfunction in animal models of PD via lowering pro-inflammatory cytokine and microglial suppression. Inhibition of microglial activation leads to the decrement of dopaminergic neurodegeneration. Consequently, polydatin regulated the Akt/glycogen synthase kinase-3β (GSK-3β)/Nrf2/NF-κB signaling axis towards such effects [169]. In addition, polydatin improved cell viability and Sirt1 expression. It also decreased mitochondrial dysfunction and ROS level [170]. Moreover, it was revealed that polydatin exerted therapeutic and protective effects in animal models of PD, including decreasing dopaminergic neuronal degeneration, neural apoptosis, and improving motor function via enhancement of glucose metabolism in neurons [171]. Polydatin also has the ability to pass through the blood–brain barrier [163]. Taken together, polydatin can be a potential therapeutic agent for treatment of PD.

Neurological I/R conditions due to oxygen and glucose deprivation lead to the occurrence of neurotoxicity and apoptosis [53,172]. Production of ROS and the inflammatory process are some of the main factors that contribute to neurological damages of I/R conditions. It has been demonstrated that polydatin, through its antioxidant, anti-inflammatory, and anti-apoptotic activities, exerts neuroprotective effects [53,62,63,167,172,173,174]. It is reported that polydatin improved neurological dysfunction and reduced infarcted area in animal models through inhibition of various CAMs, ameliorating mitochondrial dysfunction and decreasing ROS and pro-inflammatory factors [173,174,175]. Moreover, polydatin causes an enhancement in behavioral scores, reducing brain edema and hemiplegia in animal models [176].

Polydatin can improve cognitive function in conditions such as dementia [167], ethanol toxicity [177], and doxorubicin-induced cognitive impairment [81] via antioxidant [81,167], anti-inflammatory, and anti-apoptotic effects in the hippocampus [167]. Upregulation of the Nrf2/ARE signaling pathway plays an essential role in this polydatin effect [81]. In addition, polydatin ameliorated memory impairment in animal models via upregulation of brain-derived neurotrophic factor (BDNF) [178].

It is reported that polydatin has protective and therapeutic effects on SCI, such as improving motor function, reducing apoptosis, and enhancing neuron and bone marrow stromal cell (BMSC) viability [24,49,179]. Mitochondrial injury plays an essential role in SCI. Polydatin also attenuates mitochondrial dysfunction via activation of the Nrf2/ARE signaling pathway [24,180]. In addition, it is revealed that activation of the Nrf2/ARE signaling pathway by polydatin administration and BMSC transplantation led to the improvement of neuronal regeneration in animal models of SCI, facilitating BMSC differentiation and reducing glial scar formation in glial cells [181]. The reduction in inflammatory factors is another mechanism associated with polydatin’s protective and therapeutic effects against SCI [49].

It is demonstrated that polydatin has therapeutic effects against intracranial hemorrhage (ICH) and its complication [164,182]. Polydatin improved neuronal function [164] and inhibited brain edema in animal models [182]. Regulation of the Nrf2/ARE signaling pathway and downstream genes are mechanisms that are associated with the antioxidant activity of polydatin against ICH [164]. In addition, increasing levels of excitatory amino acids by polydatin may be associated with the protective and therapeutic effects of polydatin against ICH [182].

As mentioned in Table 4, activation of Nrf2-related signaling is one of the primary mechanisms involved in polydatin’s therapeutic effects against neurological diseases such as ICH, PD, and cognitive impairment. Moreover, the anti-apoptotic effects play essential roles in this field. Polydatin can be a suitable choice in the treatment of neurological diseases (Table 4).

Therapeutic targets of polydatin against cancer, cardiovascular diseases, diabetes, gastric/hepatic failure, and infection are presented in Figure 2.

### 3.8. Effects on Renal System

It has been shown that polydatin exerts therapeutic and protective effects against renal I/R injury, acute kidney injury (AKI), renal fibrosis, sepsis shock, hyperuricemia, and diabetic nephropathy through antioxidant, anti-apoptotic, and anti-inflammatory pathways [65,183,184,185,186,187].

Studies demonstrated that polydatin can decrease uric acid levels and creatinine in serum and urine [46,188,189] via the inhibition of XOD [52] in animal models of hyperuricemia. In addition, polydatin attenuated oxidative stress induced by inflammation and urates [52]. Polydatin can also ameliorate renal inflammation by activating the AMPK/Sirt1 pathway [46,189]. Consequently, polydatin showed positive effects against AKI in animal models through different mechanisms such as enhancing mitochondrial function (176) and inhibiting inflammatory and oxidative pathways [186,190]. Polydatin also decreased blood urea nitrogen (BUN) and serum creatinine in AKI animal models [186,191]. Improvement of mitophagy, inhibition of mitochondrial dysfunction [187,191], and NLRP3 inflammasome activation are related to protective effects of polydatin against AKI [187].

Oxidative stress and autophagy imbalance are two important factors associated with high-fructose-induced nephropathy. It has been depicted that polydatin activating the Nrf2 pathway leads to the attenuation of autophagy imbalance in a mTORC1-dependent manner [192]. In addition, activation of the Sirt1/Nrf2/ARE signaling pathway exerts protective effects against diabetic nephropathy. By activating this pathway, polydatin inhibited upregulation of fibronectin and TGF-β1 induced by advanced glycation end-products [65].

In conclusion, polydatin exerts therapeutic/protective effects in the treatment of renal diseases, primarily via the activation of the Nrf2 signaling pathway and inhibition of NF-κB/NLRP3 inflammasome activity (Table 5).

### 3.9. Effects on Respiratory System

Inflammatory and oxidative processes are associated with respiratory diseases such as asthma and radiation-induced lung injury. Polydatin shows protective and therapeutic effects against such diseases through antioxidant, anti-inflammatory, and anti-fibrotic mechanisms [193,194,195]. Moreover, it is shown that polydatin exerts protective and therapeutic effects against fetal conditions such as pulmonary fibrosis [196], acute respiratory distress syndrome (ARSD) [26], and lung I/R injury [197,198].

In various animal models of lung injury, polydatin showed protective and therapeutic effects such as reducing mortality, pathological alternation [151,199], lung water content [199], and formation of fibrotic tissue [193]. It also improved the ratio of arterial oxygen partial pressure (PaO2 in mmHg) to fractional inspired oxygen (PaO2/FIO2) [199]. In burning-induced lung injury models, polydatin reduced the thickness of alveolar walls, neutrophil accumulation, microvascular hyper-permeability, and polymorphonuclear leukocytes (PMNs) in bronchoalveolar lavage fluid (BAFL). In addition, polydatin reduced wet-to-dry lung weight (W/D) ratio and improved histopathological lung damage [200]. Results of radiation-induced and LPS-induced lung injury models demonstrated that polydatin exerts such effects through upregulation of HO-1 [151], club cell secretory protein (CCSP) [54,169,201], Sirt3 expression, and inhibition of epithelial–mesenchymal transition (EMT) [193]. In addition, polydatin caused anti-inflammatory effects that lead to decreases in W/D, MPO, and the number of neutrophils via regulation of the TLR4/MyD88/NF-κB signaling pathway in animal models [202]. Studies showed that polydatin exerts therapeutic effects against asthma through antioxidant and anti-inflammatory pathways [194,195]. Polydatin reduced inflammatory factors, oxidative and nitrosative stresses, fibroblasts levels, and histopathological factors via the activation of Nrf2, and reduced ROS and TGF-β levels [195]. In addition, it is revealed that the attenuation of surfactant-d (SP-D) and urocortin (UCN) is another mechanism that is associated with the anti-asthma effects of polydatin [194].

Taken together, polydatin can be a potential agent in the treatment of asthma, lung injuries, and other conditions that can lead to lung fibrosis. Anti-inflammatory, anti-apoptotic, and antioxidant activities of polydatin are obtained through decreasing PLA2 and NF-κB activity, increasing Nrf2-related signaling activity, and improving antioxidant capacities (Table 6).

### 3.10. Effects on Rheumatoid Diseases

Systematic lupus erythematosus and rheumatoid arthritis are two autoimmune diseases that can be present in various organs and systems such as the cardiovascular system, kidneys, joints, and eyes. Inflammatory and oxidative processes have essential roles in the pathogenesis of such autoimmune disorders [56,206,207]. It is revealed that polydatin has therapeutic effects against rheumatoid arthritis and its progression [56,206,208]. In vitro and in vivo studies showed that polydatin ameliorated arthritis severity and neutrophil extracellular trap (NET) accumulation in the ankles of mice [208]. In addition, it reduced matrix metalloproteinase-9 (MMP-9) [206] and ankle/hind paw diameter [56]. Regulation of the IL-6/STAT3/IL-17/NF-κB pathway is attributed to polydatin’s effects against rheumatoid arthritis [56].

The NET formation is also associated with systematic lupus erythematous pathogenesis. It is shown that polydatin inhibits ROS-mediated NET formation and NET release in vitro and in vivo. Polydatin reduced lupus markers as much as chemical drugs such as mycophenolate mofetil and cyclophosphamide. Polydatin also reduced proteinuria, circulating autoantibody levels, and the immune complex in the kidney [207].

To conclude, polydatin can be a potential compound in treating rheumatoid diseases. Lowering NET formation and decreasing inflammatory cytokines are two main mechanisms involved in such effects (Table 7).

### 3.11. Effects on the Skeletal System

Osteoporosis is one of the most common diseases among the elderly population of societies. Because of its complications such as hip fracture, it can impose much cost on the healthcare system [209]. Studies showed that polydatin exerts therapeutic and protective effects through various mechanisms in the field of osteoporosis [209,210,211], such as activation of tafazzin (TAZ) [210] and regulation of osteoprotegerin, the receptor activator of NF-κB ligand (RANKL), and B-catenin [209]. Moreover, polydatin improved osteogenic differentiation, migration of hBMSCs, and protection of the bone matrix [210,211,212]. In animal models of osteoporosis, polydatin increased ALP serum levels, calcium, ferrous, W/D, and osteoprotegerin [209]. In addition to osteoporosis, polydatin showed therapeutic effects on other skeletal diseases, including intervertebral disc degeneration and ankylosing spondylitis [30,213]. Endplate chondrocyte apoptosis is associated with the degeneration of the cartilaginous endplate (CEP) that leads to intervertebral disc degeneration [30]. Polydatin ameliorated mitochondrial injury and oxidative stress, leading to protective effects on CEP and discs against degeneration via upregulation of Nrf2 [30,214]. In addition, activation of Nrf2 caused protective effects on nucleus pulposus cells. Nucleus pulposus cells play an essential role in the inhibition/degradation of the extracellular matrix [214].

Increasing the proliferation of fibroblasts is a crucial factor in manifesting ankylosing spondylitis. In vitro studies revealed that polydatin can increase fibroblast apoptosis via upregulation of active caspase-3 expression and Bax and decreasing Bcl2. It also increased autophagy mechanisms and therapeutic effects on ankylosing spondylitis [213].

Taken together, we can consider polydatin as an efficacious agent in the treatment of osteoporosis, ankylosing spondylitis, and intervertebral disc degeneration; therefore, it could be an interesting subject for further studies in this field (Table 8).

### 3.12. Effects on Women’s Health

Endometriosis targets the female population with a prevalence of 10% [215]. There is a connection between endometriosis and inflammatory processes [31]. Mast cells that are close to nerve fibers in endometrial lesions play an essential role in regulating inflammatory processes [215]. Mast cell activation leads to the aggravation of pain and hyperalgesia in endometriosis [216]. In different studies, the effectiveness of palmitoylethanolamide (PEA)/polydatin combination was evaluated on endometriosis. PEA showed anti-inflammatory and inhibitory effects on mast cells. On the other hand, polydatin is known for its antioxidant and anti-inflammatory effects in different diseases [216]. Results in animal models of endometriosis showed that PEA/polydatin combination reduced cyst diameter, mast cell number, uterus impairment, inflammatory factors, and inflammatory cell infiltration [31,217]. In addition, clinical studies showed that PEA/polydatin combination improved chronic pelvic pain, dysmenorrhea, deep dyspareunia, and dyschezia [215,216,218]. It also improved life quality and psychological condition [218]. In addition to endometriosis, polydatin showed therapeutic and protective effects against hemorrhagic shock during pregnancy in vivo. It also improved survival time and microcirculation in animal models [219]. Table 9 shows the main benefits of polydatin on women’s health.

Therapeutic targets of polydatin against kidney injury, neurodegeneration, respiratory dysfunction, skeletal problems, and women’s disorders are presented in Figure 3.

### 3.13. Miscellaneous Effects

Polydatin is reported to have protective and therapeutic effects on other diseases and pathological conditions, including allergies [221,222,223], benign prostatic hyperplasia (BPH) [224], anxiety [225], Graves’ orbitopathy [226], pain [227], and age-related diseases [228]. Moreover, it is revealed that polydatin exerts protective effects in skin-related problems [229,230], such as facilitating epidermal growth factor receptor (EGFR) and inhibiting cutaneous side effects [230]. Studies showed that polydatin reduced pigmentation and histamine serum levels in passive cutaneous anaphylaxis and passive systemic anaphylaxis, respectively [221]. Mast cell stabilization [221,222] and decreasing inflammatory cytokines [221] are associated with such effects. In addition, polydatin attenuated food allergies via mast cell stabilization, reducing immunoglobulin E (IgE) production and improving integrity of the mucosal barrier [223]. Using polydatin in combination with PEA reduced dihydrotestosterone production and prostate weight through anti-apoptotic and anti-inflammatory mechanisms in animal models of BPH [224].

As previously mentioned, polydatin exhibited analgesic effects on endometriosis, chronic pelvic pain, and dysmenorrhea [215,216,218]. In addition, polydatin ameliorated osteoarthritis [227], cystitis/bladder pain syndrome [231], and diabetic neuropathy [232]. Polydatin reduced inflammatory factors and matrix-degrading protease in osteoarthritis through anti-inflammatory and chondroprotective effects [227]. Activation of the Nrf2 pathway is related to therapeutic and protective effects of polydatin on osteoarthritis [227] and diabetic neuropathy [232]. Polydatin also demonstrated protective effects on sciatic neurons and Schwann cells in animal models of diabetic neuropathy via inhibition of Keap 1, and activation of Nrf2/glyoxalase 1 [232].

## 4. Clinical Trials on Polydatin

Therapeutic and protective effects of polydatin have been evaluated in different clinical trials involving disorders such as chronic pelvic pain, inflammatory bowel syndrome (IBS), liver disease, and EGFR TKI-related rashes. As previously described, the effectiveness of polydatin on chronic pelvic pain related to endometriosis has been elucidated. Loi et al. assessed the effects of polydatin against chronic pelvic pain related to endometriosis in thirty symptomatic women desiring pregnancy. Patients administered 600 mg ultramicronized PEA (um-PEA) twice a day for 10 days followed by 400/40 mg co-micronized PEA/polydatin twice a day for 80 days demonstrated protective responses. The um-PEA and PEA/polydatin regimen improved pain symptoms, psychological condition, and the quality of life effectively. In addition, no adverse events were reported by this regimen, so it can be considered a potential regimen in the therapy of endometriosis symptoms [218]. Soave et al. showed that taking PEA/polydatin 400/40 mg twice a day for 90 days decreased pelvic pain and dysmenorrhea/dyspareunia. It also improved the quality of life in the 24 patients with suspected endometriosis suffering from severe pelvic pain who were involved in this study [233]. In addition, results of a review study showed that administration of PEA/polydatin 400 mg/40 mg twice a day exerted positive effects against endometriotic chronic pelvic pain, including improving chronic pelvic pain, dysmenorrhea, and deep dyspareunia [215].

Moreover, the assessment of PEA/polydatin effects against IBS on 54 patients with IBS and 12 healthy people showed potential approaches toward pain management in the IBS patients. Consequently, PEA/polydatin 200 mg/20 mg twice a day for 12 weeks reduced the fatty acid amide oleoylethanolamide. It also increased the expression of cannabinoid receptor 2, thereby significantly decreasing abdominal pain severity [234].

The protective and therapeutic effects of polydatin were investigated in 20 alcoholic patients hospitalized for rehabilitative therapy. They were given one of two oral nutritional supplements containing vitamin C or vitamin C/polydatin. It was shown that the later regiment reduced levels of AST, ALT, and lipid peroxidation in patients. Moreover, the polydatin-receiving group improved cognitive performance. Taken together, polydatin administration can be considered as a protective and therapeutic strategy in liver disease and cognitive impairment in alcoholic patients [235]. In another study, dietary supplements including polydatin and precursors for the endogenous synthesis of glutathione in healthy people are reported to be more effective on the redox status than dietary supplements containing precursors for the endogenous synthesis of glutathione alone. In the first supplement group, levels of vitamins C, E, and A are increased more than in the second group. Neopterin levels also decreased more effectively in the first group [236].

Fuggetta et al. showed that topical administration of 1.5% polydatin-based cream twice a day lowered the incidence of grade ≥2 skin toxicities in patients with mutated non-small-cell lung cancer (NSCLC) taking afatinib (40 mg/day) [230]. In addition, administration of 1.5% polydatin cream in patients with papulopustular adverse effects improved cutaneous adverse effects. Plydatin 0.8% also prevented the development of cutaneous adverse effects in patients without papulopustular adverse effects, and was used twice a day for 6 months [237].

Therefore, polydatin’s efficacy has been proven in clinical trials. Polydatin can be a good choice in the management of chronic pelvic pain, IBS, liver disease, and EGFR TKI-related rashes. Moreover, the therapeutic effects of polydatin can be considered as rational subjects for clinical trials (Table 10).

## 5. Novel Delivery Systems of Polydatin

Various novel drug delivery systems, including nanoparticles [238], liposomes [239], micelles [240], quantum dots [241], and polymeric nanocapsules [242], are designed to enhance polydatin pharmacodynamics and pharmacokinetics [243].

ZnO#ZnS quantum dot (QD) heterojunctions (QDHJs) have been reported to lower free concentration of stilbenes such as polydatin and their pharmacological effects through increasing the affinity of polydatin for common bovine plasma proteins [241]. Polydatin-loaded chitosan nanoparticles (polydatin-CSNPs) showed better efficacy than free polydatin against type 2 diabetes [244], diabetic nephropathy [245], and diabetic cardiomyopathy [238]. Polydatin-CSNPs also showed a prolonged profile release [238,244]. In animal models, polydatin-CSNPs exerted better attenuating effects against diabetic cardiomyopathy than metformin [238]. Moreover, it has been demonstrated that polydatin-loaded chitosan nanoparticles (polydatin-CSNPs) attenuated diabetic liver damage through different mechanisms, including anti-inflammatory and antioxidant effects, regulating glucose transporter 2 expression, and influencing carbohydrate metabolism enzymes. Polydatin-CSNPs had higher bioavailability than naïve polydatin, which also showed a prolonged release pattern. Therefore, polydatin-CSNPs showed more efficacy than naïve polydatin [243]. In addition to polydatin-CSNPs, the effects of a polydatin-loaded micelle (polydatin-MC) were assessed against liver fibrosis. Polydatin-MC is designed based on the ROS and pH dual-sensitive block polymer PEG-P (PBEM-*co*-DPA). The micelle enhanced the biocompatibility of polydatin, and it also improved drug release in the liver in response to the fibrotic microenvironment. Interestingly, blank micelles exerted anti-inflammatory effects by improving hepatic ROS consumption and enhancing polydatin’s therapeutic effects [240]. Polydatin-loaded liposome is another drug delivery system designed to improve polydatin’s release profile. This drug delivery system improved the oral bioavailability of polydatin and prolonged drug circulation time remarkably [239].

Polydatin-loaded polymeric nanocapsules are reported to have the same therapeutic effects as naïve polydatin against anti-inflammatory and antioxidant activities on LPS-induced changes in hippocampal organotypic cultures. On the other hand, the encapsulation procedure enhanced polydatin bioavailability and stability [242]. Polydatin-encapsulated poly [lactic-co-glycolic acid] (polydatin-PLGA-NPs) showed preventive effects against oral squamous cell carcinoma (OSCC) in animal models. It also inhibited lipid peroxidation, oxidative stress, and modulated phase I and phase II of the detoxification system [246].

Polydatin gold nanoparticles (polydatin-AuNP) significantly reduced dense tumor cells in Ehrlich ascites carcinoma animal models. Polydatin-AuNPs also exerted protective effects against doxorubicin’s cardiovascular side effects [247]. Moreover, polydatin-modified Au@AgNPs exerted positive effects in topical wound healing. Administration of topical polydatin-modified Au@AgNPs led to higher migration of keratinocytes [248].

To sum up, novel drug delivery systems have improved the pharmacokinetics and pharmacodynamics properties of polydatin. Improvements in bioavailability, biocompatibility, and efficacy are the main beneficial effects of polydatin’s novel drug delivery systems.

## 6. Conclusions

The complex pathophysiological mechanisms behind chronic diseases urge the need for providing novel multi-targeting agents with higher efficacy/bioavailability and lower side effects. The plant kingdom is now a great source of natural secondary metabolites with the potential of targeting multiple dysregulated pathways in diseases. Polydatin is a glucoside derivative of resveratrol possessing better bioavailability and higher health-promoting/disease-modifying activities. The anti-inflammatory, antioxidant, and apoptotic-modifying properties of polydatin provide protective effects against inflammatory, oxidative stress, and apoptotic responses. Prevailing evidence reveals the biological activities and health-promoting effects of polydatin in various conditions, including cancer, cardiovascular diseases, diabetes, gastrointestinal/hepatic diseases, neurodegeneration, and infectious conditions, as well as protective roles in the respiratory system, the renal system, rheumatoid diseases, the skeletal system, and women’s health. Polydatin applies such effects through multiple mediators (Figure 4). As such, the broad therapeutic potential of polydatin urges the need for finding potential sources to promote the application of polydatin in industrial, commercial, and research sectors. Of polydatin sources, *Reynoutria japonica* is a useful highly invasive plant and a rich source in the pharmaceutical industry [7]. However, to better improve the bioavailability and efficacy of polydatin in clinical trials, investigating appropriate delivery systems in order to solve the pharmacokinetic limitation is of great importance.

Today, pan-assay interference compounds (PAINS) have been proposed as a threat to identifying the bioactivity of natural compounds, including compounds such as resveratrol [249]. PAINS cause disruption in the membrane by reducing the membrane dipole potential and, as a result, can affect intracellular signaling pathways. PAINS-type behavior can affect intracellular signaling pathways in a non-specific way. The cell-based data of some phytochemicals might be a product of such effects and not caused by specific binding to therapeutic targets. Recent studies have shown that C-glucosylation of polyphenol compounds causes the PAINS effect, while *O*-glucosylation has no effect on this behavior [250]. As per previous data [250], polydatin, an *O*-glucosylated derivative of resveratrol, is expected to present the same concerns as the aglycone with regards to inducing non-specific reductions in the membrane dipole potential. Therefore, in order to assess if *C*-glucosylation of resveratrol is able to preserve the promising bioactivities of polydatin described in this review while addressing the mentioned PAINS-related issue, the biological evaluation of this *C*-glucosyl analog of polydatin is of utmost importance. All biological activities, health benefits, and pharmacological mechanisms of action of polydatin have been highlighted in the current study. A further area of research should include extensive pre-clinical studies to reveal the precise pharmacological mechanisms of polydatin followed by well-controlled clinical trials towards therapeutic potentials. Such reports will help to provide more biological applications of polydatin and other stilbenoids in the prevention, management, and treatment of several diseases.

## Figures and Tables

**Figure 1 molecules-27-06474-f001:**
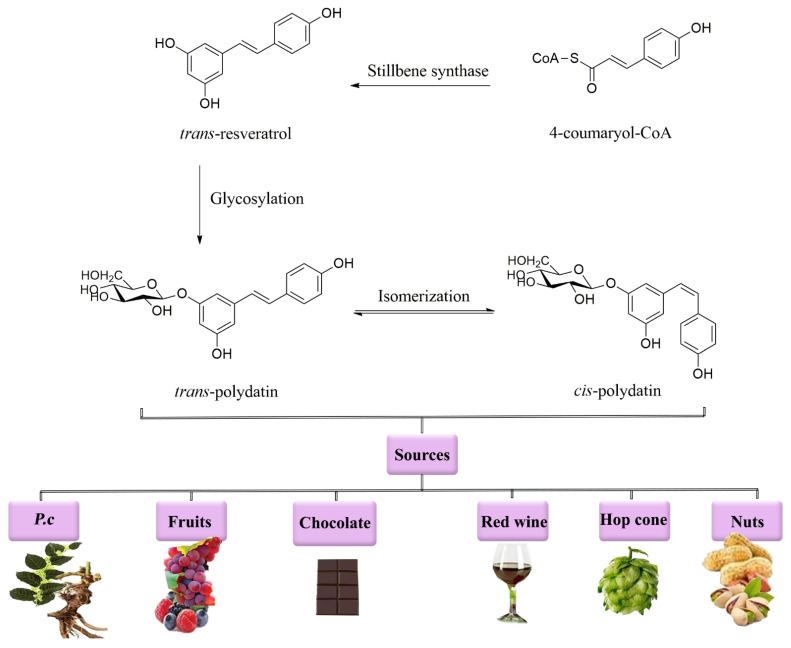
Chemical structure and sources of polydatin. *P.c*: *Polygonum cuspidatum*.

**Figure 2 molecules-27-06474-f002:**
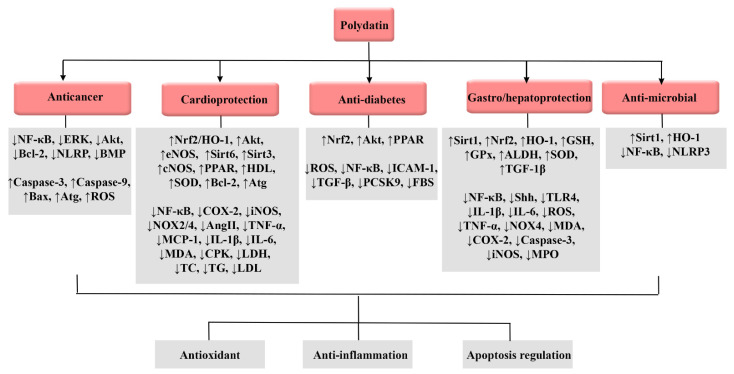
Therapeutic targets of polydatin against cancer, cardiovascular diseases, diabetes, gastric/hepatic failure, and infection. ADH: alcohol dehydrogenase, Akt: protein kinase B, ALDH: aldehyde dehydrogenase, ALT: alanine transaminase, AST: aspartate transaminase, Atg: autophagy-related gene, Bax: Bcl-2-associated x, Bcl-2: B-cell lymphoma 2, BMP: bone morphogenetic protein, CAT: catalase, cNOS: constitutive nitric oxide synthase, COX-2: cyclooxygenase-2, CPK: creatine phosphokinase, eNOS: endothelial nitric oxide synthase, ERK: extracellular signal-regulated kinase, FBS: fasting blood sugar, GSH: glutathione, GPx: glutathione peroxidase, HDL: high-density lipoprotein, HO-1: heme oxygenase-1, ICAM-1: intercellular adhesion molecule-1, IL: interleukin, iNOS: inducible nitric oxide synthase, LDH: lactate dehydrogenase, LDL: low-density lipoprotein, MCP-1: monocyte chemoattractant protein-1, MDA: malondialdehyde, MPO: myeloperoxidase, NF-κB: nuclear factor-kappa B, NO: nitric oxide, NOX: NADPH oxidase 4, NLRP3: nucleotide-binding domain (NOD)-like receptor protein 3, NOS: nitric oxide synthase, Nrf2: nuclear factor erythroid 2-related factor 2, PCSK9: proprotein convertase subtilisin/kexin type-9, PPAR: peroxisome proliferator-activated receptor, PBEF: pre-B-cell colony-enhancing factor, ROS: reactive oxygen species, Shh: Sonic hedgehog, Sirt: sirtuin, SOD: superoxide dismutase, TC: total cholesterol, TFEB: transcription factor EB, TG: triglyceride, TGF-β: transforming growth factor-beta1, TLR4: Toll-like receptor 4, TNF-α: tumor necrosis factor-α.

**Figure 3 molecules-27-06474-f003:**
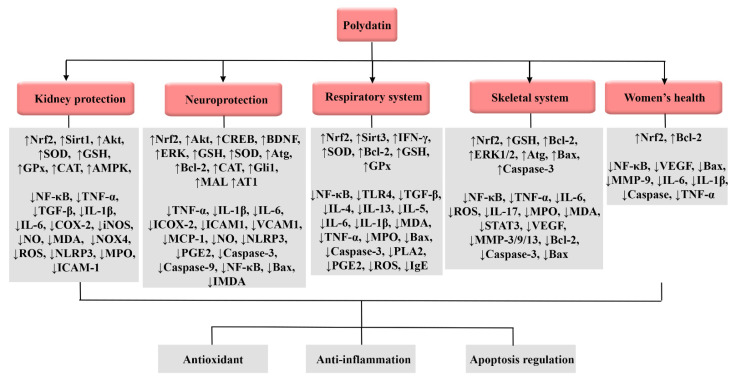
Therapeutic targets of polydatin against kidney injury, neurodegeneration, respiratory dysfunction, skeletal problems, and women’s disorders. Akt: protein kinase B, AMPK: adenosine 5′-monophosphate-activated protein kinase, Atg: autophagy-related gene, Bax: Bcl-2-associated x, Bcl-2: B-cell lymphoma 2, BDNF: brain-derived neurotrophic factor, CAT: catalase, COX-2: cyclooxygenase-2, CREB: cAMP response element-binding proteins, ERK: extracellular signal-regulated kinase, GSH: glutathione, GPx: glutathione peroxidase, ICAM-1: intercellular adhesion molecule-1, IgE: immunoglobulin E, IL: interleukin, INF-γ: interferon-γ, iNOS: inducible nitric oxide, MALAT1: metastasis-associated lung adenocarcinoma transcript 1, MCP-1: monocyte chemoattractant protein-1, MDA: malonaldehyde, MMP: matrix metalloprotease, MPO: myeloperoxidase, NO: nitric oxide, NF-κB: nuclear factor-kappa B, NOX4: NADPH oxidase 4, NLRP3: nucleotide-binding domain (NOD)-like receptor protein 3, Nrf2: nuclear factor erythroid 2-related factor 2, PGE2: prostaglandin E2, PLA2: phospholipase A2, RANKL: receptor activator of nuclear factor-kappa Β ligand, ROS: reactive oxygen species, SOD: superoxide dismutase, STAT: signal transducer and activator of transcription, TGF-β1: transforming growth factor-beta1, TLR4: Toll-like receptor 4, TNF-α: tumor necrosis factor-α, CAM-1: vascular cell adhesion molecule-1, VEGF: vascular endothelial growth factor.

**Figure 4 molecules-27-06474-f004:**
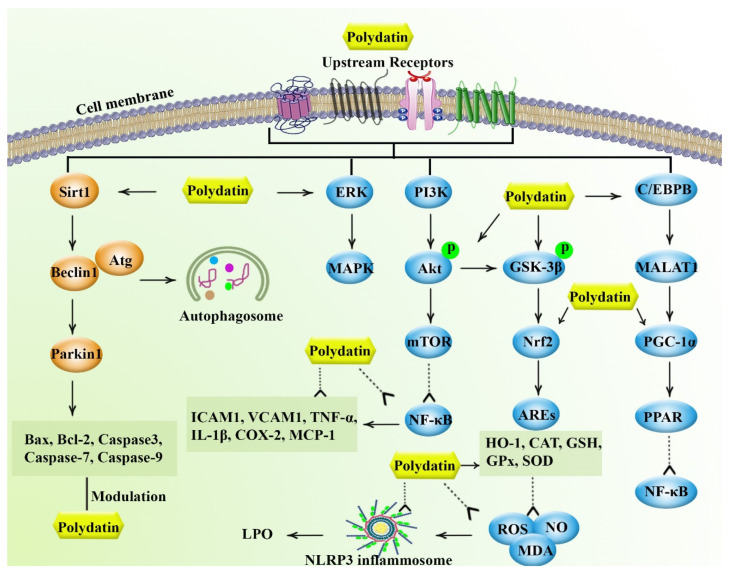
Major pharmacological mechanisms and therapeutic targets of polydatin towards biological activities and health benefits. ┬: inhibition or downregulation, →: increase or upregulation, Akt: protein kinase B, AREs: antioxidant response elements, Bax: Bcl-2-associated x, Bcl-2: B-cell lymphoma 2, C/EBPB: CCAAT/enhancer-binding proteinβ, CAT: catalase, COX-2: cyclooxygenase-2, Creb: cAMP response element-binding proteins, ERK: extracellular signal-regulated kinase, GSH: glutathione, GPx: glutathione peroxidase, GSK-3β: glycogen synthase kinase-3β, HO-1: heme oxygenase-1, IL-1β: interleukin-1β, ICAM-1: intercellular adhesion molecule-1, MALAT1: metastasis-associated lung adenocarcinoma transcript 1, MAPK: mitogen-activated protein kinases, MDA: malondialdehyde, MMP: mitochondrial membrane potential, mTOR: mammalian target of rapamycin, MCP-1: monocyte chemoattractant protein-1, NLRP3: nucleotide-binding domain (NOD)-like receptor protein 3, NF-κB: nuclear factor-kappa B, NO: nitric oxide, Nrf2: nuclear factor erythroid 2-related factor 2, PGC-1α: peroxisome proliferator-activated receptor γ coactivator 1α, PGE2: prostaglandin E-2, PPARγ: peroxisome proliferator-activated receptor γ, ROS: reactive oxygen species, Sirt1: sirtuin1, SOD: superoxide dismutase, TLR: Toll-like receptor, TNF-α: tumor necrosis factor-α, VCAM-1: vascular cell adhesion molecule-1.

**Table 1 molecules-27-06474-t001:** Anticancer effects of polydatin.

Study Type	Biological Effects	Cell Lines, Animal Types	Doses	References
In vivo, in vitro	⇅miR-382/PD-L1 axis activity, ┬cell proliferation, ↑cleaved caspase-3 and cleaved caspase-9 expression	Mice, colorectal cancer cell line (Caco-2, HCT 116, HEK293)	150 mg/kg/day for 16 days, injection into tumor, 50, 100, 150 μM	[88]
In vivo, in vitro	↑BMP signaling pathway activity, radiosensitivity, ┬proliferation	Mice, colon tumor cell line (CT26, HCT116)	25 mg/kg i.p. once a week for 4 weeks, 0, 40, 60, 80, 100, and 120 μM	[89]
In vitro	↑Bax and cyclin A expression, ↓cyclin D1 and Bc1-2 expression, ┬growth, ↑arresting cells in S phase	Human acute monocytic leukemia cell line (THP-1)	10, 20, 40, 60, 80, 100, 120, 140, 160 μM for 24 and 48 h	[90]
In vitro	↑Bcl-2-associated x expression, ↓cyclin D1 and cyclin B1 expression	Leukemia cell line (MOLT-4)	0.5, 1, 2, 4, 10, or 20 μM	[86]
In vivo, in vitro	┬PDGF/Akt signaling pathway activity, cell proliferation, ↓Bcl-2 expression, ↑Bax expression	Mice, human laryngeal cancer cell line (AMC-HN-8, HeLa Hep-2)	50 mg/kg 3 times a week for 3 weeks, 2, 4, 6 μM for 24, 48, 72 h	[91]
In vitro	↑ROS level, ER stress, cytosolic cytochrome c level, ↓mitochondrial cytochrome C level, Akt phosphorylation	Human nasopharyngeal carcinoma cell line (CNE), cervical carcinoma cell line (HeLa), hepatoma cell line (SMMC-7721), epidermal carcinoma cell line (A-431)	-	[92]
In vitro	↓phosphorylated Her-2 and EGFR levels, ERK expression, secretion of VEGF	Human ovarian adenocarcinoma cell line (SKOV-3, OVCAR-8)	5, 10, 50, 100 μM for 6 days	[93]
In vitro	↓phosphor-NF-κB p6 expression, activation of NF-κB pathway, ┬NLRP3 inflammasome activation, proliferation and migration of NSCLC cells	NSCLC cell line (A549, H1299)	25, 50, 100 μM for 24 h	[94]
In vitro	↑Bax expression, ↓Bcl-2 and cyclin D1 expression, ⇅mTOR pathway activity	Lung cancer cell line (A549)	2, 4, 6, 8 μM for 24 and 48 h	[95]
In vivo, in vitro	┬EGFR-Akt/ERK1/2/STAT3-SOX2/Snail signaling pathway activity, ┬proliferation, migration, invasion, stemness, ↑Bax expression, ↓Bcl-2 and cyclin D1 expression	Mice, lung cancer cell lines (NCI-H1975, A549), breast cancer cell lines (MDA-MB-231, MCF-7), cervical cancer cell line (HeLa), ovarian cancer cell line (SKOV-3), liver cancer cell line (SMMC-7721), nasopharyngeal cancer cell line (CNE-1), leukemia cell line (HL-60, K562)	50 mg/kg/day for 14 days i.p. injection, 0–8 μM for 24 and 48 h	[96]
In vitro	↓phosphorylation of mTOR and p70s6k	Human multiple myeloma cell line (RPMI 8226)	0, 5, 10, 25, 50, 100, 200, and 400 μM for 24 and 48 h	[98]
In vitro	┬PI3K/Akt/mTOR signaling pathway activity and downstream gene expression	Human cervical cancer cell line (HeLa cells)	50, 100, 150 μM	[99]
In vivo, in vitro	┬c-Myc expression, ⇅cell cycle-related protein expression (p21, p27, CDK2, CDK4, Cyclin D1, Cyclin E1, EMT, markers (E-cadherin, N-cadherin, Snail and Slug)), ┬proliferation and metastasis	Mice, human cervical cancer cell lines (CaSki, C33A), human embryonic renal cell line (293FT)	100 mg/kg/day injection for 3 weeks, 0.1, 10, 100, 500 μM for 24 and 48 h	[100]
In vivo, in vitro	┬G6PD activity, ↑ROS level, ER stress	Mice, head, and neck squamous cell carcinoma (HNSCC), breast cancer cell line (MCF-7)	100 mg/kg 3 times a week for 30 days, 2–100 μM for 24 or 48 h	[1]
In vitro	┬G6PD activity, ↑autophagosomes formation	Breast cell lines (MCF-7)	10, 20, 30 μM for 24 h	[21]
In vitro	↓phosphorylated CREB level, cyclin D1 expression, ↑S-phase cell cycle arrest	Breast cancer cell lines (MDA-MB-231, MCF-7)	0–8 μM for 24 and 48 h	[101]
In vivo, in vitro	┬ROS/PI3K/Akt/HIF-1α/HK2 signaling axis activity	Mice, breast cell lines (4T1, MCF-7)	100 mg/kg i.p. injection every other day for 3 weeks, 2 μM	[102]
In vitro	┬Akt activation, ⇅Ki67, p21, cyclin A, cyclin E, CDK2, MMP-2, MMP-9, tissue metalloproteinase-1, PPAR 1, caspase-3, p-glycoprotein 1, lung resistance-related protein 1, growth arrest, and DNA damage-45α, glutathione S-transferase π and heat shock protein 27 expression, ↑chemosensitivity to paclitaxel	Human osteosarcoma cell lines (U-2OS, MG-63)	0–400 μM for 24, 48, 72 h	[103]
In vitro	┬STAT3 signaling pathway activity, ↑Atg12, Atg14, BECN, and PIC3K3 expression, autophagic cell death	Human normal osteoblast cell (hFOB1.19), human osteosarcoma cell line (MG-63)	0–160 μM for 12–72 h	[104]
In vitro	┬β-catenin signaling pathway activity, ↓cell proliferation, ↑Bax expression, ↓Bcl-2 expression, caspase-3 activity	Osteosarcoma cell lines (143B, MG63)	0, 1, 10, 30, 100 μM for 24, 48, 72 h	[105]
In vivo, in vitro	┬TUG1/Akt signaling pathway activity	Mice, osteosarcoma cell lines (Saos-2, MG-63), normal osteoblast cell line (hFOB), doxorubicin-resistant cell lines (Saos-2/Dox, MG- 63/Dox)	150 mg/Kg/day i.p. injection for 20 days, 50, 100, 150, 200, 250 μM 24, 48, 72 h	[106]
In vivo, in vitro	↑caspase-3 activity, caspase-3, caspase-9 and Bax expression, TUNEL activity, ↓expression of Bcl-2, Ki-67, ┬Wnt/beta-catenin signaling activity	Mice, normal liver cell lines (HL-7702, HCC), liver cancer cell lines (HepG2, SMMC-7721)	25, 50, 100 mg/kg/day i.p. injection for 20 days, 1, 3, 10, 30, 100 μM	[107]
In vitro	┬Akt/STAT3–FOXO1 signaling pathway activity	Human hepatocellular carcinoma cell line (HCCLM3), normal hepatic cell line LO2	0–800 μM for 24–72 h	[108]

↓: decrease, ↑: increase, ┬: inhibit, ⇅: regulate, μM: micromole, Akt: protein kinase B, Atg: autophagy-related gene, Bax,: Bcl-2-associated x, Bcl-2: B-cell lymphoma 2, BMP: bone morphogenetic protein, c-Myc: cellular Myc, cAMP: cyclic adenosine monophosphate, CDK: cyclin-dependent kinase, CREB: cAMP response element-binding proteins, E-cadherin: epithelial cadherin, EGFR: epidermal growth factor receptor, EMT: epithelial mesenchymal transition, ER: endoplasmic reticulum, ERK: extracellular signal-regulated kinase, FOXO1: forkhead box protein O1, G6PD: glucose-6-phosphate dehydrogenase, HIF-1α: hypoxia-inducible factor, HK2: hexokinase 2, i.p.: intraperitoneal, kg: kilogram, mg: milligram, mTOR: mammalian target of rapamycin, N-cadherin: neural cadherin, NF-κB: nuclear factor-kappa B, NLRP3: nucleotide-binding domain (NOD)-like receptor protein 3, NSCLC: non-small-cell lung cancer, PI3K: phosphoinositide 3-kinases, ROS: reactive oxygen species, SOX2: sex-determining region Y, STAT: signal transducer and activator of transcription, TUG1: taurine-upregulated gene 1, VEGF: vascular endothelial growth factor.

**Table 2 molecules-27-06474-t002:** Effects of polydatin on cardiovascular activity.

Study Type	Biological Effects	Cell Lines, Animal Types	Doses	References
In vivo, in vitro	↑Nrf2/HO-1 signaling activity, ↓ROS levels, hypoxia, AMI-induced myocardial damage	Rat, rat embryonic cardiomyocyte cells	100 mg/kg/day i.p. injection for 30 days, 0, 10, 100, 250, 500 μM for 12 h	[114]
In vitro	↑eNOS/Sirt1 pathway activity, ↓proliferation of VSMCs, ROS	vascular smooth muscle cells (VSMCs)	10, 50,100 μM for 24 h	[115]
In vivo	⇅NO, Ang II and ET expression, ┬protein kinase C activity	Rat	5, 10, 20 mg/kg every other day i.p. injection for 3 weeks	[116]
In vivo	↓ICAM-1, V-CAM, TNF-α, IL-1β, iNOS, PARP, and NF-κB expression, Bax and FAS ligand activity, vascular damage	Mice	30 mg/kg orally for 14 days	[117]
In vivo, in vitro	↑Sirt6-mediated autophagy and Bcl-2 expression, ↓serum TNF-α, IL-1β, IL-6, caspase-3, and Bax expression	Rat, cardiomyoblast cell line (H9c2)	30 mg/kg i.p. injection once, 100 μM for 30 min	[118]
In vivo	↑reverse cholesterol transport	Mice	50, 100 mg/kg/day orally for 12 weeks	[127]
In vivo	↓ROS levels, ${\mathrm{Ca}}^{2+}$ spark-mediated sarcoplasmic reticulum leak	Rat	10 mg/kg i.v. injection once	[135]
In vivo	↓size of cardiomyocyte, aldosterone, TNF-α, Ang II, and endothelin-1 expression, blood pressure	Rat, mice	60, 120 mg/kg/day orally for 30 days	[119]
In vivo	↓renin–angiotensin–aldosterone system activity, ┬peroxidation	Rat	25, 50, 100 mg/kg/day orally for 7 weeks	[120]
In vivo, in vitro	┬NADPH oxidase activity, superoxide production, ↓Ang II-mediated cardiac hypertrophy	Rat, ventricle cells of neonatal rats	50 mg/kg/day orally for 7 days, 5–50 μM for 60 min	[121]
In vitro, in vivo	↑Sirt3 expression, autophagy flux	Mice, cardiomyocytes of neonates, and adult mice	7.5 mg/kg/day injection for 28 days	[122]
In vivo	↓NOX4, NOX2, NF-κB expression, NADPH oxidase activity, ROS level, inflammatory cytokines	Rat, embryonic rat cardiac cell line H9c2	100 mg/kg/day orally for 8 weeks	[35]
In vivo	↓myocardial superoxide generation, MDA level, $\mathrm{gp}91^{phox}$ and iNOS expression, NO metabolite levels, nitrotyrosine, ↑Notch1/Hes1-Pten/Akt signaling pathway activity, eNOS phosphorylation	Rat	20 mg/kg/day orally for 4 days	[36,66,123]
In vivo	↑PPARβ activation, ↓NF-κB p65, COX-2, and iNOS expression	Mice	50, 100 mg/kg/day	[124]
In vivo	↑LDL-C/HDL-C, TC/HDL-C ratios, ↓serum TC, TG, LDL-C, hepatic TG levels	Hamsters	25, 50 and 100 mg/kg/day orally for 15 days	[125]
In vivo	↓TC, TG, LDL-C, TC/HDL-C	Rabbit	25, 50, 100 mg/kg/day orally for 3 weeks	[126]
In vitro	↑eNOS/Sirt1 pathway activity, Kip1/p27 expression, ↓VSMCs proliferation, ROS production	VSMCs of the mouse thoracic aorta	10, 50,100 μM for 24 h	[115]
In vivo, in vitro	↓PBEF-inducing cholesterol deposition in macrophages	Mice, peritoneal macrophages of mouse	100 mg/kg/day i.p. injection for 12 weeks, 50, 100, 200 μg/mL for 2 h	[128]
In vivo, in vitro	↓TC, TG, LDL-C, PPARγ expression, MCP-1 and TNF-α expression, ↑HDL, leptin expression	Mice, mouse 3T3-L1 preadipocytes	100 mg/kg/d orally for 4 weeks, 20 μM for 2 days	[130]
In vitro	⇅PPARγ signaling pathways	Mouse peritoneal macrophages	8.9 μg/mL	[131]
In vivo	↑PKC-$K_{\mathrm{ATP}}$-dependent signaling activity, ↓CPK and LDH leakage, ⇅SOD and MDA content	Rat	20 μg/kg, i.v. once	[71]
In vitro	↑Bcl-2 expression, ↓Bax expression, ↓I/R-induced apoptosis	Rat heart	50 μM for 10 min	[132]
In vivo, in vitro	┬RAS and the downstream ROCK pathway activity	Rat, ventricular myocytes of rats	50 mg/kg/day orally for 8 days, 30, 50 μM for 1 h	[22]
In vivo, in vitro	↑SOD activity, NO, NOS activity, cNOS, ↓MDA content	Rat, hearts of rats	0.2 mL/100 g body weight (0.1% polydatin solution) i.v. injection before model establishment, 0.05 μM 15 min before model establishment	[133]
In vivo, in vitro	↑Sirt3 activity, ↑mitochondrial biogenesis, upregulation of autophagy	Mice, ventricular cardiomyocytes	7.5 mg/kg/day i.p. injection for 10 days, 10 μM	[2]

↓: decrease, ↑: increase, ┬: inhibit, ⇅: regulate, μg: microgram, μM: micromole, Akt: protein kinase B, Ang II: angiotensin II, Bax: Bcl-2-associated x, Bcl-2: B-cell lymphoma 2, cNOS: constitutive nitric oxide synthase, COX-2: cyclooxygenase-2, CPK: creatine phosphokinase, ${\mathrm{Ca}}^{2+}$: calcium ion, eNOS: endothelial nitric oxide synthase, ET: endothelin, $\mathrm{gp}91^{phox}$: glycoprotein P 91 phagocytic oxidase, HDL-C: high-density lipoprotein cholesterol, HO-1: heme oxygenase-1, i.p.: intraperitoneal, i.v.: intravenous, ICAM-1: intercellular adhesion molecule-1, IL-1β: interlukin-1β, IL-6: interleukin-6, iNOS: inducible nitric oxide synthase, kg: kilogram, Kip1/p27: a cyclin–CDK inhibitor, LDL-C: low-density lipoprotein cholesterol, LDH: lactate dehydrogenase, MDA: malondialdehyde, mg: milligram, MCP-1: monocyte chemoattractant protein-1, NADPH: nicotinamide adenine dinucleotide phosphate, NF-κB: nuclear factor-kappa β, NO: nitric oxide, NOS: nitric oxide synthase, NOX4: NADPH oxidase 4, NOX2: NADPH oxidase 2, Notch1: Notch homolog 1 translocation-associated, Nrf2: nuclear factor erythroid 2-related factor 2, PARP: poly (ADP-ribose) polymerase, PBEF: pre-B-cell colony-enhancing factor, PKC: protein kinase C, PPARβ: peroxisome proliferator-activated receptor β, PPARγ: peroxisome proliferator-activated receptor γ, RAS: renin–angiotensin system, ROCK: rho kinase, ROS: reactive oxygen species, Sirt1: sirtuin1, Sirt6: sirtunin6, SOD: superoxide dismutase, TC: total cholesterol, TG: triglyceride, TNF-α: tumor necrosis factor-α, V-CAM: vascular cell adhesion molecule, VSMCs: vascular smooth muscle cells.

**Table 3 molecules-27-06474-t003:** Hepatoprotective effects of polydatin.

Study Type	Biological Effects	Animal Types, Cell Lines	Doses	References
In vitro, in vivo	↓CYP2Y3, CYP3A65, HMGCRa, HMGCRb, FASN, lipid and ethanol metabolism, hepatic fat accumulation, oxidative stress, DNA damage↑CHOP and GADD45aa expression	Zebrafish larvae	6.25, 12.5, 25 μg/mL for 48 h	[153]
In vivo	⇅TLR4 and NF-κB activity, ↓TNF-α, IL-1β, IL-6, CYP2E1, and ROS levels, ↑Nrf2/HO-1, SOD activity, GSH-Px, CAT, ADH, and ALDH activity	Rat	25, 50, 100 mg/kg/day orally for 7 days	[48,155]
In vivo	↓lipid peroxidation, TNF-α expression, lipogenesis	Rat	30, 90 mg/kg/day orally for 8 weeks	[156]
In vivo, in vitro	┬mTOR signaling activity, ↑TFEB expression, autophagic flux	Mice, human hepatocyte cell line (LO2)	100 mg/kg every other day orally for 4 weeks, 24 μM for 24 h	[154]
In vivo, in vitro	↓NOX4 enzymes, oxidative stress, CD68, FAS, SREBP-1c, LDH, cleaved caspase-3, annexin V, ALT, AST levels, ┬TLR-4/NF-κB p65 signaling pathway activity,	Mice, human hepatocellular carcinoma cell line (HepG2)	5 mg/kg, every other day, i.p. injection for 4 weeks, 5, 10, 20 μM for 24 h	[157]
In vivo, in vitro	↑Sirt1/Nrf2 pathway activity, ↓serum aminotransferase levels, hepatic pathological damage	Mice, human hepatocyte cell line (L02)	100, 200, 400 mg/kg/day orally for 7 days, 50 μM for 24 h	[158]
In vivo	↓4-HNE production, NOX4 expression	Mice	5 mg/kg i.p. injection	[159]
In vivo	↓AST, ALT, MDA, TNF-α, IL-1β, COX-2, iNOs, NF-κB, ↑GSH content, GST, SOD, CAT, and GSH-Px activity, hepatic TGF-b1 expression	Mice	25, 50, 100 mg/kg/day orally for 5 days	[70]
In vivo	┬NF-κB activation, ↓TNF-α expression, MPO activity, ICAM-l and ECAM-l expression, caspase-3 activity	Mice	10, 30,100 mg/kg i.p. injection	[55,161]

↓: decrease, ↑: increase, ┬: inhibit, ⇅: regulate, μg: microgram, μM: micromole, μM: microgram, 4-HNE: 4-hydroxy-2-nonenal, ADH: alcohol dehydrogenase, ALDH: aldehyde dehydrogenase, ALT: alanine transaminase, AST: aspartate transaminase, CAT: catalase, CD68: cluster of differentiation 68, COX-2: cyclooxygenase-2, CYP: cytochrome P450, DNA: deoxyribonucleic acid, ECAM-1: epithelial cell adhesion molecule-1, FASN: fatty acid synthase, GADD45αa: growth arrest and DNA-damage-inducible alpha a, GSH: glutathione, GSH-Px: glutathione peroxidase, GST: glutathione transferase, HMGCRa: hydroxymethylglutaryl-CoA reductase a, HMGCRb: hydroxymethylglutaryl-CoA reductase b, HO-1: heme oxygenase-1, i.p.: intraperitoneal, ICAM-1: intercellular adhesion molecule 1, IL-1β: interleukin-1β, IL-6: interleukin-6, iNOs: inducible nitric oxide synthase, kg: kilogram, LDH: lactate dehydrogenase, MDA: malondialdehyde, mg: milligram, MPO: myeloperoxidase, mTOR: mammalian target of rapamycin, NF-κB: nuclear factor-kappa B, ROS: reactive oxygen species, NOX4: NADPH oxidase 4, Nrf2: nuclear factor erythroid 2-related factor 2, Sirt1: sirtuin1, SOD: superoxide dismutase, SREBP-1c: sterol regulatory element-binding protein 1c, TFEB: transcription factor EB, TGF-β1: transforming growth factor-beta1, TNF-α: tumor necrosis factor-α, TLR4: Toll-like receptor 4.

**Table 4 molecules-27-06474-t004:** Neuroprotective effects of polydatin.

Study Type	Biological Effects	Cell lines, Animal Types	Doses	References
In vivo	↓MDA and activated caspase-3 levels, ↑GSH levels, SOD activity, p-Akt level, motor functions, ⇅TRX and ATP levels	Rat	20, 40, 80 mg/kg/day orally for 5 weeks	[163]
In vivo, in vitro	↑autophagy-related gene 5 (Atg5) activity, mTOR/Ulk-involved autophagy, ↓PGC1β/mfn2-involved mitochondrial fusion, ⇅autophagic processes, mitochondrial fusion	Flies, human neuroblastoma cell (SH-SY5Y)	2 mM orally for 25 days, 1–500 μM for 6 h	[170]
In vivo, in vitro	⇅Akt/GSK3β-Nrf2/NF-κB signaling axis activity	Rat, murine microglia cell line (BV-2 cells)	25, 50, 100 mg/kg/day orally for 32 days, 100, 200, 400 μM for 1 h	[169]
In vitro	↓caspase-3/7 activity, ↑Bcl-2 expression, ERK1/2, and ERK5 activity	Human dopaminergic neuroblastoma cell line (SH-SY5Y)	10, 20, 30 μM for 1 h	[9]
In vivo	glucose metabolism enhancement	Mice	20, 80 mg/kg/day orally for 10 days	[171]
In vivo	⇅Nrf2/ARE pathway and downstream gene expression, ↑SOD, GSSG, and GSH levels, ↓NO and MDA levels	Rat	50 mg/kg orally	[164]
In vivo, in vitro	↓MDA production,↑SOD and CAT activity	Rat, rat cortical neuron	12.5, 25, 50 mg/kg/day orally for 30 days, 12.5, 50 μg/mL for 1 h	[167]
In vivo	↓Bax expression, caspase-9 activity, caspase-3 activity, ROS levels, infarcted volume, mitochondrial dysfunction, ↑Bcl-2 expression	Rat	30 mg/kg i.v. injection	[173]
In vivo	↓infarction volume, neurobehavioral deficits, neuronal apoptosis, ROS level, p38 mitogen-activated protein kinase activity, c-Jun N-terminal kinase activity, ↑Nrf2/HO-1/TRX activity	Rat	30 mg/kg i.p. injection twice	[62,167,174]
In vivo	↑Gli1, Ptch1, and SOD1 expression, ↓NF-κB expression, BBB permeability, infarcted volume, brain water content	Rat	25, 50 mg/kg i.p. injection once or for 2 days	[63]
In vivo	↓brain edema	Rat	12.5, 50 mg/kg i.p. injection once	[176]
In vivo	↑Nrf2 activity, MDA and GSH levels, ↓hippocampal apoptosis, TUNEL-positive cells, cleaved caspase-3, cleaved caspase-9, NF-κB, TNF-α, PGE-2, and COX-2 levels	Rat	50 mg/kg/day orally for 3 or 4 weeks	[81]
In vivo, in vitro	↓NO, IL-1β, IL-6, and TNF-α levels, NLRP3 inflammasome activation, ↑motor function	Rat, BV2 mouse microglia	20, 40 mg/kg i.p. injection once, 1, 2, 4 μM for 24 h	[49]
In vivo, in vitro	↑Nrf2/ARE pathway activity, ↓mitochondrial dysfunction	Mice, spinal cord motor neurons	30 mg/kg/day orally for 9 days, 3.75, 7.5, 15, 30, or 60 μM	[180]
In vitro	⇅Nrf 2/ARE pathway activity	BMSCs	3, 10, 30 μM for 2 h	[179]
In vivo, in vitro	↑Nrf2 pathway activity	Mice, BMSCs	20 mg/kg/day gastrically perfused for 5 days, 3, 10, 30 μM	[181]
In vivo	↑BDNF expression	Rat	10 mg/kg/day orally for 10 days	[178]
In vivo	↓ICAM-1, VCAM-1, E-selectin, L-selectin, and Integrins levels, ↓neurological deficits, volume of brain infarction	Rat	7.5, 15, 30 mg/kg i.v. injection once	[175]
In vivo, in vitro	↑MALAT1/CREB/PGC-1α/PPARγ signaling pathway activity, BBB integrity, ↓cerebral infarcted volume, TNF-α, IL-6, IL-1β, COX-2, ICAM-1, VCAM-1, and MCP-1 expression	Rat, human embryonic kidney cell line (HEK-293T), human umbilical vein endothelial cell line (HUVEC)	30 mg/kg i.v. injection once, 20 μM for 24 h	[53,172]
In vivo	↑intracellular ATP, lysosomal stability, ↓mitochondrial swelling, vasoresponsiveness, arteriolar smooth muscle cell hyperpolarization, K_ATP_ channel activity	Rat	15, 30, and 45 mg/100 g body weight	[166]

↓: decrease, ↑: increase, ⇅: regulate, μg: microgram, μM: micromole, Akt: protein kinase B, ARE: antioxidant response element, Atg5: autophagy-related gene 5, ATP: adenosine triphosphate, Bax: Bcl-2-associated x, BBB: blood–brain barrier, Bcl-2: B-cell lymphoma 2, BDNF: brain-derived neurotrophic factor, CAT: catalase, COX-2: cyclooxygenase-2, CREB: cAMP-response element-binding protein, E-selectin: CD62 antigen-like family member E, ERK: extracellular signal-regulated kinase, HO-1: heme oxygenase-1, i.v.: intravenous, i.p.: intraperitoneal, IL-1β: interleukin-1β, IL-6: interleukin-6, L-selectin: CD62L, K: potassium, kg: kilogram, g: gram, Gli1: glioma-associated oncogene homologue 1, GSH: glutathione, GSK3β: glycogen synthase kinase-3β, GSSG: glutathione disulfide, ICAM-1: intercellular adhesion molecule 1, MALAT1: metastasis-associated lung adenocarcinoma transcript 1, MCP-1: monocyte chemoattractant protein-1, MDA: malondialdehyde, mTOR: mammalian target of rapamycin, mfn2: mitofusin 2, NF-κB: nuclear factor-kappa B, NLRP3: nucleotide-binding domain (NOD)-like receptor protein 3, NO: nitric oxide, Nrf2: nuclear factor erythroid 2-related factor 2, PGC-1α: peroxisome proliferator-activated receptor γ coactivator 1α, PGC1β: proliferator-activated receptor γ coactivator 1β, PGE-2: prostaglandin E-2, PPARγ: peroxisome proliferator-activated receptor *γ*, Ptch1: patched 1, ROS: reactive oxygen species, SOD: superoxide dismutase, Ulk: Unc-51-like kinase, TNF-α: tumor necrosis factor-α, TRX: thioredoxin, TUNEL: Terminal deoxynucleotidyl transferase (TdT) dUTP nick-end labeling, VCAM-1: vascular cell adhesion molecule.

**Table 5 molecules-27-06474-t005:** Effect of polydatin on the renal system.

Study Type	Biological Effects	Cell Lines, Animal Types	Doses	References
In vivo	⇅renal expression of mURAT1, mGLUT9, mAGCG2, mOAT1, mOCT1, mOCTN1, mOCTN2, mUMOD, ↑antihyperuricemic, nephroprotective effect	Mice	20, 40 mg/kg/day orally for 7 days	[183]
In vivo	↑PI3K/Akt pathway activity, SOD, GST, GSH-Px, and CAT activity, GSH level, ↓TNF-α, IL-1, COX-2, iNOS, PGE-2, NO, and MDA levels	Mice	10,20,40 mg/kg/day i.p. injection for 6 days	[10,50,184]
In vivo	↓NF-κB p65, COX-2, and iNOS expression, ┬TNF-α, PGE-2, and IL-1β production	Mice	12.5, 25.0, 50.0 mg/kg/day orally for 4 weeks	[52]
In vivo	↑AMPK activity, Sirt1 expression, ┬NF-κB/NLRP3 inflammasome activity	Rats	25, 50 mg/kg/day orally for 7 days	[46,189]
In vivo, in vitro	↑Cx32 expression, K48-linked polyubiquitination activity, ↓NOX4, ROS, FN, and ICAM-1 levels	Mice, primary GMCs of rat	100 mg/k/day orally 6 days a week for 12 weeks, 5, 10, 20, 40, 80, 160 μM for 24 h	[185]
In vivo	┬NF-κB signaling, ↑Nrf2 signaling pathway activity, ↓TNF-α, IL-1β, and IL-6 expression, MPO activity, MDA content	Mice	20, 40, 80 mg/kg i.p. injection once	[186]
In vivo, in vitro	↑Sirt1/Nrf2/ARE pathway activity, ↓ROS, FN, and TGF-β1 levels	Rats, GMCs of rat	150 mg/kg/day orally for 12 weeks, 5, 10, 20 μM	[65]
In vivo, in vitro	↓AMPK/p38 MAPK signaling pathway activity, autophagy imbalance, ↑Nrf2 signaling pathway activity	Rat, conditionally immortalized human podocytes (HPCs)	7.5, 15, 30 mg/kg/day orally for 6 weeks, 5, 10, 20 μM for 24 h	[192]
In vivo, in vitro	↑Sirt1–p53 expression, ↓mitochondrial dysfunction	Rat, human renal proximal tubular epithelial cell line (HK-2)	30 mg/kg infusion for 2 h, various concentrations for 2 h and 50 μM for 48 h	[191]
In vivo	↓IL-6 level, lipid peroxide, lysosomal instability, mitochondrial swelling, ↑ATP level, mitochondrial membrane potential	Rats	30 mg/kg i.v. injection three times (6, 12, and 18 h after model establishment)	[190]
In vivo	↑Sirt1 activity, Parkin-dependent mitophagy, ↓NLRP3 inflammasome activity, mitochondrial dysfunction	mice	30 mg/kg i.v. injection once	[187]

↓: decrease, ↑: increase, ┬: inhibit, ⇅: regulate, μg: microgram, μM: micromole, Akt: protein kinase B, AMPK: adenosine 5′-monophosphate-activated protein kinase, ARE: antioxidant response element, ATP: adenosine triphosphate, CAT: catalase, COX-2: cyclooxygenase-2, FN: fibronectin, GMCs: glomerular mesenchymal cells, GSH: glutathione, GSH-Px: glutathione peroxidase, GST: glutathione transferase, ICAM-1: intercellular adhesion molecule 1, i.p.: intraperitoneal, i.v.: intravenous, IL-1β: interleukin-1β, IL-6: interleukin-6, iNOS: inducible nitric oxide synthase, kg: kilogram, MAPK: mitogen-activated protein kinase, MDA: malondialdehyde, mg: milligram, MPO: myeloperoxidase, NF-κB: nuclear factor-kappa B, NLRP3: nucleotide-binding domain (NOD)-like receptor protein 3, NO: nitric oxide, NOX4: NADPH oxidase 4, Nrf2: Nuclear factor erythroid 2-related factor 2, PGE-2: prostaglandin E-2, PI3K: phosphatidylinositol 3-kinase, ROS: reactive oxygen species, Sirt1: sirtuin1, TGF-β1: transforming growth factor-beta1, TNF-α: tumor necrosis factor-α.

**Table 6 molecules-27-06474-t006:** Effect of polydatin on respiratory system.

Study Type	Biological Effects	Cell Lines, Animal Types	Doses	References
In vivo, in vitro	┬TGF-β1/Smad3 signaling pathway activity, epithelial–mesenchymal transition, ↓IL-4 and IL-3 levels, ↑Sirt3 expression, Nrf2 and PGC1a levels, INF-γ activity	Mice, human bronchial epithelial cell line (BEAS-2B)	100 mg/kg/day i.p. injection for 31 days	[193]
In vivo	↓MDA level, W/D, ultrastructure injuries, ↑SOD content	Rabbit	-	[198]
In vivo	↓TLR4 and NF-κB expression, ┬inflammatory mediators	Rabbit	-	[197]
In vivo	↓TNF-α, IL-1β, and IL-6 levels, MPO activity, total cells, PMNs in BALF, Bcl-xl expression, ↑Bax and caspase-3 activity	Rat	15, 30, 45 mg/kg via caudal vein once	[199]
In vivo, in vitro	↑mitophagy via Parkin, Bcl-2 expression, ↓mitochondrial-dependent apoptosis, Bax expression, cytochrome c release, caspase-3 activity	Mice, human bronchial epithelial cell line (BEAS 2B cells)	45 mg/kg once, 50 μM for 6 h	[26]
In vivo	┬PLA2	Rat	1 mg/kg i.v. injection once	[203]
In vitro, in vivo	┬TGF-β/Smad/ERK pathway activity, ↓α-smooth muscle actin, collagen I, TNF-α, IL-6, IL-13, MPO, and MDA expression, ↑epithelial cell cadherin expression, SOD activity	Rat	10, 40, and 160 mg/kg/day for 28 days	[196]
In vivo	↓PLA2 activity, hydroxyproline, PGE-2, LTC4, and TGF-β1 levels in BALF	Rat	10, 20, 40 mg/kg intraperitoneal injection once before model establishment	[204]
In vivo, in vitro	┬TGF-β1 activity, ↓ROS activity, ↑Nrf2 signaling pathway activity	Mice, human lung epithelial cell line (BEAS-2B)	100 mg/kg at days 0, 7, 14, 100 μM for up to 48 h	[195]
In vivo	⇅SP-D and UCN expression, ↓serum IgE, IL-4, IL-13, IL-5, MDA, TNF-α, IFN-γ, iNOS, and eosinophil levels, ↑SOD activity, GSH level	Rats	200 mg/kg/day orally for 14 days	[194]
In vitro	↑GSH-Px and SOD content, ↓apoptosis, MDA content, activation-related proteins of the NLRP3 inflammasome, TNF-α, TGF-β, IL-1β, IL-6	Embryonic lung fibroblast MRC-5	12.5, 25, 50, 100, 200, 400 μmol/L for 24 h	[205]
In vitro, in vivo	↑CCSP expression, ┬PLA2 activation	Rats, human bronchial epithelia cells (BEAS-2B cells)	1, 5, 10, 30 mg/kg i.v. injection once, 0.5 m mol/L for 4 h	[54,169,201]
In vivo	↓TUNEL-positive lung cells, TNF-α, IL-1, and IL-6 levels, lung MPO activity, ┬Bax upregulation, caspase-3 activity, Bcl-xl downregulation	Rat	45 mg/kg i.v. injection once	[200]
In vitro, in vivo	┬TLR4/MyD88/NF-κB signaling pathway activity, W/D, MPO, neutrophil number, ↓IL-6, IL-1β, TNF-α, IL-8 levels	Mice, human bronchial epithelial cell line (BEAS-2B cells)	20, 80 mg/kg i.p. injection once, 2 μM, 4 μM, 8 μM for 2 h	[202]
In vivo	┬NF-κB activity, ↓TNF-α and IL-6 levels, lung COX-2 and iNOS expression	Mice	15, 45, 100 mg/kg i.p. injection once	[151]

↓: decrease, ↑: increase, ┬: inhibit, ⇅: regulate, μg: microgram, μM: micromole, BALF: bronchoalveolar lavage fluid, Bax: Bcl-2-associated x, Bcl: B-cell lymphoma 2, CCSP: club cell secretory protein, COX-2: cyclooxygenase-2, ERK: extracellular signal-regulated kinase, GSH: glutathione, GSH-Px: glutathione peroxidase, i.p.: intraperitoneal, i.v.: intravenous, IgE: immunoglobulin E, INF-γ: interferon-γ, iNOS: inducible nitric oxide synthase, IL-3: interleukin-3, IL-4: interleukin-4, IL-13: interleukin-13, kg: kilogram, LTC4: leukotriene C4, MDA: malondialdehyde, mg: milligram, MPO: myeloperoxidase, MyD88: myeloid differentiation factor 88, NF-κB: nuclear factor-κB, NLRP3: nucleotide-binding domain (NOD)-like receptor protein 3, Nrf2: nuclear factor erythroid 2-related factor 2, PGC1a: peroxisome proliferator-activated receptor γ coactivator 1α, PLA2: phospholipase A2, ROS: reactive oxygen species, Sirt3: sirtuin3, Smad3: mothers against decapentaplegic homolog 3, SOD: superoxide dismutase, SP-D: surfactant-d, TGF-β1: transforming growth factor-β1, TLR4: Toll-like receptor 4, TNF-α: tumor necrosis factor-α, TUNEL: terminal deoxynucleotidyl transferase (TdT) dUTP nick-end labeling, UCN: urocortin, W/D: wet-to-dry.

**Table 7 molecules-27-06474-t007:** Effects of polydatin on rheumatoid diseases.

Study Type	Biological Effects	Cell Lines, Animal Types	Doses	References
In vivo, in vitro	┬NET formation	Mice, neutrophils isolated from RA patients and mice	45 mg/kg/day i.p. injection for 24 days, 0, 50, 75, 100, and 125 μg/mL	[208]
In vivo	↓TNF-α, IL-6, IL-17, and MMP levels, MPO activity, MDA level, ↓RANKL and STAT3 expression, ┬VEGF, NF-κB, ↑GSH content	Rats	200 mg/kg/day orally for 21 days	[56]
In vivo	↓IL-1β, TNF-α, MDA, and SOD expression, ↑Bcl-2/Bax pathway expression, MMP-9 activity, ┬caspase-3/9 activity	Mice	15, 30, 45 mg/kg i.d. injection for 24 h	[206]
In vivo, in vitro	┬ROS-mediated NET formation	Mice, neutrophils from systematic lupus erythematous patients and healthy people	45 mg/kg/day i.p. injection for 8 or 16 weeks, 50, 75, 100, 125, and 150 μg/mL	[207]

↓: decrease, ↑: increase, ┬: inhibit, μg: microgram, μM: micromole, Bax: Bcl-2-associated x, Bcl-2: B-cell lymphoma 2, GSH: glutathione, i.d: intradermal, IL-1β: interleukin-1β, IL-6: interleukin-6, IL-17: interleukin-17, i.p.: intraperitoneal, kg: kilogram, MDA: malondialdehyde, mg: milligram, MMP: matrix metalloprotease, NET: neutrophil extracellular trap, NF-κB: nuclear factor-κB, RANKL: receptor activator of nuclear factor-kappa Β ligand, ROS: reactive oxygen species, SOD: superoxide dismutase, STAT3: signal transducer and activator of transcription, TNF-α: tumor necrosis factor-α, VEGF: vascular endothelial growth factor.

**Table 8 molecules-27-06474-t008:** Effects of polydatin on skeletal system.

Study Type	Biological Effects	Cell Lines, Animal Types	Doses	References
In vivo, in vitro	↑Nrf2 signaling pathway activity, Parkin expression	Rats, human endplate chondrocytes	50 mg/kg/day orally for 4 weeks, 200 μM for 2 h	[30]
In vivo, in vitro	⇅BMP2–Wnt/β-catenin, ↑TAZ, mRNAs RUNX2, osteopontin, DLX5, β-catenin, and osteocalcin expression	Mice, hBMSCs	3 mg/kg every 2 days i.p. injection for 12 weeks, 0, 30, 10, 100 μM for 1, 2, 3, 7, 14 days	[210]
In vitro	↑ERK ½ activity	Mouse BMSCs	5, 10, 15, 30, 100 μM for 24 h	[212]
In vitro	↑active caspase-3, Bax, LC3II, Beclin 1, and Atg5 expression, ↓Bcl2 expression	Fibroblasts of patients with AS	0, 0.33, 1, 3, 10 μM for 24, 48, or 72 h	[213]
In vivo, In vitro	↑aggrecan and collagen II levels, Nrf2 signaling pathway activity, ↓TNF-α, p53, p16, MMP-3, MMP-13, and ADAMTS-4 expression	Rats, nucleus pulposus cells of rats	50 mg/kg/day orally for 4 weeks, 0, 200, 400 μmol/L for 24 h	[214]
In vivo, in vitro	⇅osteoprotegerin, RANKL, β-catenin expression	Mice, mouse bone marrow stromal cell line ST2	10, 20, 40, mg/kg/day i.p. injection for 12 weeks, 20, 40 μg/mL for 48 h	[209]

↓: decrease, ↑: increase, ⇅: regulate, μg: microgram, μM: micromole, ADAMTS-4: a disintegrin and metalloproteinase with thrombospondin motif 4, Atg5: autophagy-related gene, Bax: Bcl-2-associated x, Bcl2: B-cell lymphoma 2, BMP2: bone morphogenetic protein 2, DLX5: distal-less homeobox 5, ERK: extracellular signal-regulated kinase, hBMSCs: human bone marrow stromal cells, i.p.: intraperitoneal, kg: kilogram, LC3II: microtubule-associated protein 1A/1B-light chain 3 ll, mg: milligram, MMP: matrix metalloprotease, Nrf2: nuclear factor erythroid 2-related factor 2, RANKL: receptor activator of nuclear factor-kappa Β ligand, RUNX2: runt-related transcription factor 2, TAZ: tafazzin, TNF-α: tumor necrosis factor-α.

**Table 9 molecules-27-06474-t009:** Effects of polydatin on women’s health.

Study Type	Biological Effects	Cell Lines, Animal Types	Doses	References
In vivo	↓VEGF, nerve growth factor, ICAM and MMP-9 expression, lymphocyte accumulation, peroxynitrite formation, PARP activation, IkBa phosphorylation, NF-κB translocation	Rats	10 mg/kg/day orally for 14 days	[217]
In vivo	↑capillary perfusion, functional capillary density, survival time	Rabbits	single bolus infusion 4 mL/kg	[219]
In vivo	↓NF-κB activity, IL-6, IL-1β, 8-OHdG, 4-HNE, IL-6, *γ*-H2AX, Bax, cleaved-caspase-3, IL-6, caspase-1, phosphorylated p6 levels	Mice	50, 100 mg/kg/day orally for 28 days	[220]
In vivo	┬NF-κB pathway activity, ↑Nrf2 signaling pathway activity, ↓TNF-α, IL-1β, and IL-6 expression	Mice	20, 40, 80 mg/kg i.p. injection once	[31]

↓: decrease, ↑: increase, ┬: inhibit, μg: microgram, μM: micromole, γ-H2AX: γ-H2A histone family member X, 4-HNE: 4-hydroxy-2-nonenal, 8-OHdG: 8-hydroxydeoxyguanosine, Bax: Bcl-2-associated x, i.p.: intraperitoneal, ICAM: intercellular adhesion molecule, IkBa: inhibitory kappa B kinases a, IL-1β: interleukin-1β, IL-6: interleukin-6, kg: kilogram, mg: milligram, mL: milliliter, MMP-9: matrix metalloprotease-9, NF-κB: nuclear factor-κB Nrf2: nuclear factor erythroid 2-related factor 2, PAR: poly-ADP ribose, VEGF: vascular endothelial growth factor.

**Table 10 molecules-27-06474-t010:** Clinical trials on polydatin.

Clinical Study Type	Health Benefits	Gender	Dose	Study Type	References
Women’s health	↓chronic pelvic pain, deep dyspareunia, dysmenorrhea, dyschezia, ↑life quality	Symptomatic women with endometriosis desiring pregnancy	600 mg um-PEA orally twice a day for 10 days followed by 400/40 mg m(PEA/polydatin) orally twice a day for another 80 days	single-arm, non-randomized, open-label clinical study	[218]
Gastrointestinal	↓abdominal pain severity	IBS patients and healthy people	200/20 mg PEA/polydatin twice a day for 12 weeks	randomized, double-blind, placebo-controlled, multi-center clinical study	[234]
Liver disease	↓AST and ALT levels, lipid peroxidation, ↓cognitive impairment	Alcoholic patients	40 mg orally twice a day for 2 weeks	A pilot study	[235]
Antioxidant and anti-inflammatory activity	↓oxidized species levels, neopterin levels,↑GSH level, other thiol species, vitamins C, E, and A	Healthy people	35 mg orally twice a day for 8 weeks	randomized clinical study	[236]
Skin	↓pruritus, skin manifestation such as papulopustular rash,↑quality of life	Cancer patients taking anti-EGFR treatment regimen with or without cutaneous adverse events	Cream (1.5% or 0.8%) twice a day topically for 6 months	prospective pilot study	[237]
Skin	↓incidence of rash	Patients with mutated NSCLC taking afatinib	1.5% cream twice a day topically for 6.4 months	pilot study	[230]

↓: decrease, ↑: increase, ALT: alanine transaminase, AST: aspartate transaminase, EGFR: epidermal growth factor receptor, GSH: glutathione, IBS: irritable bowel syndrome, mg: milligram, NSCLC: non-small-cell lung cancer, PEA: palmitoylethanolamide.

## Data Availability

Data sharing is not applicable to this article as no new data were created or analyzed in this study.

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
