# Peer review of "Polydatin: Pharmacological Mechanisms, Therapeutic Targets, Biological Activities, and Health Benefits"

_molecules, 2022, doi:10.3390/molecules27196474_

Round 1

Reviewer 1 Report

The work of Ahmad et al., reviews the biological activities and pharmacological mechanisms of polydatin to provide new insights to researchers in order to promote the compound for future uses. The work has value, especially due to the fact that recent studies shown that polydatin presents a greater antioxidant capacity than resveratrol but doesn't receives the same attention as resveratrol, this is mainly due to Polydatin sourcing, whereas resveratrol is nowadays highly associated with grapes and wine.
Introduction
One of the main sources of polydatin (and resveratrol) industrial scale production is Reynoutria japonica, an invasive plant from the Far East, so invasive it reached even the most remote mountain areas where it unbalances the ecosystem including the agricultural potential of mountain areas, being advantaged by the climate change context [1]. This data has to be added in order to bring to knowledge the potential use of this invasive plant as it is the highest Polydatin and Resveratrol producer, also the effects of Polydatin on damaged macrophages [2] should be stated (between Lines 40-49).
2.2. Antioxidant effects
Polydatin effects on RNS (Reactive Nitrogen Species) should be stated.
2.3. Anti-apoptotic effects
Apoptotic effects should as well be mentioned, especially mitochondrial apoptosis effects in carcinoma cells [3]
Conclusions
Should state the need of promoting Polydatin in industrial, commercial and research sectors.

1. Negrea B-M, Stoilov-Linu V, Pop C-E, Deák G, Crăciun N, FăgăraÈ™ MM. Expansion of the Invasive Plant Species Reynoutria japonica Houtt in the Upper BistriÈ›a Mountain River Basin with a Calculus on the Productive Potential of a Mountain Meadow. Sustainability. 2022; 14(9):5737. https://doi.org/10.3390/su14095737 2. Liu, C., Wang, W., Zhang, K., Liu, Q., Ma, T., Tan, L., & Ma, L. (2022). Protective Effects of Polydatin from Grapes and Reynoutria japonica Houtt. on Damaged Macrophages Treated with Acetaminophen. Nutrients, 14(10), 2077. https://doi.org/10.3390/nu14102077
3. Liu, H., Zhao, S., Zhang, Y., Wu, J., Peng, H., Fan, J., & Liao, J. (2011). Reactive oxygen speciesmediated
endoplasmic reticulum stress and mitochondrial dysfunction contribute to polydatininduced
apoptosis in human nasopharyngeal carcinoma CNE cells. Journal of Cellular Biochemistry,
112.

Author Response

Reviewer #1 Overall comment: The work of Ahmad et al., reviews the biological activities and pharmacological mechanisms of polydatin to provide new insights to researchers in order to promote the compound for future uses. The work has value, especially due to the fact that recent studies shown that polydatin presents a greater antioxidant capacity than resveratrol but doesn't receives the same attention as resveratrol, this is mainly due to Polydatin sourcing, whereas resveratrol is nowadays highly associated with grapes and wine.

Response: We appreciate the respected reviewer for the positive consideration of our manuscript for publication and for describing the fact of the biological capacities of polydatin. The “point-by-point” and itemized modification has been done in light of your comments. Comment 1: Introduction One of the main sources of polydatin (and resveratrol) industrial scale production is Reynoutria japonica, an invasive plant from the Far East, so invasive it reached even the most remote mountain areas where it unbalances the ecosystem including the agricultural potential of mountain areas, being advantaged by the climate change context [1]. This data has to be added in order to bring to knowledge the potential use of this invasive plant as it is the highest. Polydatin and Resveratrol producer, also the effects of Polydatin on damaged macrophages [2] should be stated (between Lines 40-49).

Response 1: This is a critical comment which is now added to the revised manuscript (lines 29- 33; Reference 7) AND (lines 58-59; Reference 32).

Comment 2: 2.2. Antioxidant effects Polydatin effects on RNS (Reactive Nitrogen Species) should be stated.

Response 2: It is now stated in the revised manuscript (lines 131-132; References 44 and 55). Comment 3: 2.3. Anti-apoptotic effects Apoptotic effects should as well be mentioned, especially mitochondrial apoptosis effects in carcinoma cells [3].

Response 3: It is now added (lines 183-186; Reference 74).

Comment 4: Conclusions Should state the need of promoting Polydatin in industrial, commercial and research sectors.

Response 4: Thank you. It is stated in conclusion (lines 923-925). 1. Negrea B-M, Stoilov-Linu V, Pop C-E, Deák G, Crăciun N, FăgăraÈ™ MM. Expansion of the Invasive Plant Species Reynoutria japonica Houtt in the Upper BistriÈ›a Mountain River Basin with a Calculus on the Productive Potential of a Mountain Meadow. Sustainability. 2022; 14(9):5737. https://doi.org/10.3390/su14095737 2. Liu, C., Wang, W., Zhang, K., Liu, Q., Ma, T., Tan, L., & Ma, L. (2022). Protective Effects of Polydatin from Grapes and Reynoutria japonica Houtt. on Damaged Macrophages Treated with Acetaminophen. Nutrients, 14(10), 2077. https://doi.org/10.3390/nu14102077 3. Liu, H., Zhao, S., Zhang, Y., Wu, J., Peng, H., Fan, J., & Liao, J. (2011). Reactive oxygen speciesmediated endoplasmic reticulum stress and mitochondrial dysfunction contribute to polydatininduced apoptosis in human nasopharyngeal carcinoma CNE cells. Journal of Cellular Biochemistry, 112.

Reviewer 2 Report

Review on the manuscript.

Manuscript title: Polydatin: Pharmacological mechanisms, therapeutic targets, 1 biological activities, and health benefits

The authors have reviewed the modulation of pivotal signaling pathways involved in inflammation, oxidative stress, and apoptosis by Polydatin has been reviewed.  Various  biological activities including anticancer, cardioprotective, anti-diabetic, gastroprotective, hepatoprotective, neuroprotective, anti-microbial, as well as health-promoting roles on renal system, respiratory system, rheumatoid disease, skeletal system, and women health has been reviewed     to provide new insights to researchers about polydatin.

The manuscript was well written with sufficient details with respect to the biological activities of polydatin.

 In my opinion the manuscript can be accepted with the following revisions.

It possesses four leading derivatives in nature, including trans-polydatin, trans-resveratrol, cis-polydatin, and cis-resveratrol. How one could differentiate the above isomers when it is isolated. Some chemistry could be added.

The therapeutic and protective effects of polydatin mainly originated from its anti-inflammatory, antioxidant, and anti-apoptotic activities.  A few lines about how the therapeutic and protective effects of polydatin could be explained from its anti-inflammatory, antioxidant, and anti-apoptotic activities.

In Fig 1 stilbene synthase has converted 4-coumaryol-CoA in to trans-resveratrol.  In the text nowhere it has been explained.  Is it self-explanatory or any explanation is required.

By the presence of glucose group the bioavailability increases. How what is the reason behind it.

Author Response

Overall comment: The authors have reviewed the modulation of pivotal signaling pathways involved in inflammation, oxidative stress, and apoptosis by Polydatin has been reviewed.  Various biological activities including anticancer, cardioprotective, anti-diabetic, gastroprotective, hepatoprotective, neuroprotective, anti-microbial, as well as health-promoting roles on renal system, respiratory system, rheumatoid disease, skeletal system, and women health has been reviewed     to provide new insights to researchers about polydatin.

The manuscript was well written with sufficient details with respect to the biological activities of polydatin. In my opinion the manuscript can be accepted with the following revisions.

Response: We greatly express our sincere thanks to respected reviewer for the careful review and positive consideration of our manuscript towards publication. The “point-by-point” and itemized modification has been done in the light of your comments.

Comment 1: It possesses four leading derivatives in nature, including trans-polydatin, trans-resveratrol, cis-polydatin, and cis-resveratrol. How one could differentiate the above isomers when it is isolated. Some chemistry could be added.

Response 1: Thanks. Isolation methods are now briefly described (lines 44-47).

Comment 2: The therapeutic and protective effects of polydatin mainly originated from its anti-inflammatory, antioxidant, and anti-apoptotic activities.  A few lines about how the therapeutic and protective effects of polydatin could be explained from its anti-inflammatory, antioxidant, and anti-apoptotic activities.

Response 2: Thank you for the comment. Comment done (lines 67-70).

Comment 3: In Fig 1 stilbene synthase has converted 4-coumaryol-CoA in to trans-resveratrol.  In the text nowhere it has been explained.  Is it self-explanatory or any explanation is required.

Response 3: In the light of your comment, we added a brief description on the reaction, however, it was clear in the Figure 1 (lines 42-44).

Comment 4: By the presence of glucose group the bioavailability increases. How what is the reason behind it.

Response 4: Thank you. Description done on regulating the intestinal absorption and metabolism of resveratrol by adding glucose group in polydatin (lines 37-39). 

Reviewer 3 Report

Dear all,

The present review covers the bioactivity of polydatin in a variety of therapeutic contexts, highlighting the antioxidant, anti-inflammatory, and apoptosis-modulating effects of this natural compound. Despite the valuable effort made by the authors to cover a vast amount of literature focusing on this compound, I do believe that this article contains a great number of redundant ideas, together with an overall flat description of published data, resulting in reader disengagement. Tables and Figures are too descriptive and do not contribute, in my point of view, to a significant content clarification. Hence, my opinion is that the review should be considered for acceptance after major revision.

My comments are as follows:

  1. Page 1, line 33: The sentence “...easily absorbed from the intestinal tract due to the glucose group.” is incorrect. Polyphenol O-glycosides are more soluble in water than their aglycones, which is advantageous for compound dissolution and overall bioavailability, but they are actually hydrolyzed in the gut. It is mostly the aglycone that passes through the intestinal barrier into the bloodstream. Please check the following articles and correct the sentence appropriately: 

https://www.frontiersin.org/articles/10.3389/fnut.2022.857879/full 

https://www.ncbi.nlm.nih.gov/pmc/articles/PMC4352739/ 

Moreover, the fact that C-glucosides are not as easily cleaved by glucosidases should, in my opinion, be emphasized in this review, in a way that suggests that the C-glucosylation of resveratrol could potentially be applied to solve this problem. Please consider the following references as well: 

https://pubmed.ncbi.nlm.nih.gov/25199800/ 

https://www.mdpi.com/2304-8158/11/6/882 

Importantly, the C-glucoside of resveratrol is already described in the literature; however, its biological properties have not yet been elucidated. I believe that this should be highlighted in order to encourage the scientific community to investigate this compound with regard to its biological properties. Please see here the first report on C-glucosyl resveratrol and consider citing it in your discussion: https://www.nature.com/articles/s41598-021-83032-3 

  1. Page 1, line 42: “disease” should be replaced with “diseases” and “women” with “women’s”.

  1. Figure 1. “Glucosydase” or, correctly spelled “Glucosidase” is an enzyme that cleaves a glycosidic bond, it does not catalyze glucosylation reactions. Also, the glucosyl moiety is not correctly drawn: it is missing the endocyclic oxygen, and the C-OH bonds have to be parallel to those in the 6-membered ring (i.e., C-2 is correct; C-3, C-4 and C-6 are incorrect). Please correct accordingly.

  1. Page 2, line 46: Please replace “are provided” with “are disclosed” or an equivalent expression.

  1. Page 2, lines 55-56: This sentence is redundant, please consider removing it.

  1. Page 3, line 88: Please replace “comprised” with “orchestrated” or an equivalent expression.

  1. Page 4, line 166 and following lines: From here, the authors are describing the apoptotic effects of PD in cancer cell models, which makes its inclusion in the “Anti-apoptotic effects” section inappropriate. Please include this paragraph in the “Anticancer effects” section.

  1. Page 5, line 178: As per my previous comment, this sentence does not make sense. PD seems to exert anti-apoptotic activity in damaged tissues, but pro-apoptotic activity in cancer cells. I recommend revising this summary of the section.

  1. Page 5, line 185-187. This is a literal repetition of page 1, line 40. Please rephrase and/or abbreviate.

  1. Page 5, line 198: Reference 78 (Higuera-Ciapara, I.; Felix-Valenzuela, L.; Goycoolea, F., Astaxanthin: a review of its chemistry and applications. Critical reviews 1124 in food science and nutrition 2006, 46, (2), 185-196) revises the bioactivity of astaxanthin, not polydatin.

  1. Page 5, line 206: “...as well as induction of apoptosis…” this should be connected to the pro-apoptotic effects of PD in cancer cell lines, according to my suggestion in comment 7.

  1. Page 5, line 214: in “2-deoxy-D-glucose”, “D” should be written in small capitals.

  1. Table 1 and following tables: “increase” and “deacrease” needs to be further specified for the meaning of the arrows. Is it increased/decreased protein expression? Or increased/decreased activity?

  1. Page 12, line 422: Reference 136 (Zhang, X.; Liu, X. D.; Xian, Y. F.; Zhang, F.; Huang, P. Y.; Tang, Y.; Yuan, Q. J.; Lin, Z. X., Berberine enhances survival and axonal 1259 regeneration of motoneurons following spinal root avulsion and re-implantation in rats. Free radical biology & medicine 2019, 143, 1260 454-470) focuses on berberin, not PD.

  1. Page 21, line 670: “rheumatoid disease” should be replaced with “rheumatoid diseases”, plural.

  1. Page 23, line 731: “women health” should be replaced with “women’s health”.

  2. Figure 1: Two yellow PD icons side by side (affecting Sirt1 and ERK) are not needed. Please delete one of them and have both arrows coming from the same icon. Also, please see comment 18.

  1. Final general comment: Something of extreme importance that is not discussed in the article is the fact that resveratrol is a PAIN (Pan-Assay Interference Compound) that works as a membrane disruptor by reducing the membrane dipole potential, which may affect the activation/inactivation of intracellular signalling pathways in a non-specific way, when used in cell-based assays. Thus, most results presented in this review have to be safeguarded with the fact that PD presents huge concerns when regarded as a lead for pharmaceutical development. O-glycosides of known PAINS are not likely to behave differently, as shown by Rauter et al. (https://doi.org/10.1038/s41598-021-83032-3). This has to be clear throughout, or in a dedicated section, as well as in Figure 4. C-glucosyl resveratrol, on the other hand, no longer presents this PAINS issue. Again, I strongly advise the authors to develop this topic in their discussion.

Kind regards.

Author Response

Reviewer #3

Overall comment: The present review covers the bioactivity of polydatin in a variety of therapeutic contexts, highlighting the antioxidant, anti-inflammatory, and apoptosis-modulating effects of this natural compound. Despite the valuable effort made by the authors to cover a vast amount of literature focusing on this compound, I do believe that this article contains a great number of redundant ideas, together with an overall flat description of published data, resulting in reader disengagement. Tables and Figures are too descriptive and do not contribute, in my point of view, to a significant content clarification. Hence, my opinion is that the review should be considered for acceptance after major revision.

Response: We greatly express our sincere thanks to respected reviewer for the careful review and positive consideration of our manuscript towards publication. The “point-by-point” and itemized modification has been done in the light of your comments.

Comment 1: Page 1, line 33: The sentence “...easily absorbed from the intestinal tract due to the glucose group.” is incorrect. Polyphenol O-glycosides are more soluble in water than their aglycones, which is advantageous for compound dissolution and overall bioavailability, but they are actually hydrolyzed in the gut. It is mostly the aglycone that passes through the intestinal barrier into the bloodstream. Please check the following articles and correct the sentence appropriately:

https://www.frontiersin.org/articles/10.3389/fnut.2022.857879/full

https://www.ncbi.nlm.nih.gov/pmc/articles/PMC4352739/

Moreover, the fact that C-glucosides are not as easily cleaved by glucosidases should, in my opinion, be emphasized in this review, in a way that suggests that the C-glucosylation of resveratrol could potentially be applied to solve this problem. Please consider the following references as well:

https://pubmed.ncbi.nlm.nih.gov/25199800/

https://www.mdpi.com/2304-8158/11/6/882

Importantly, the C-glucoside of resveratrol is already described in the literature; however, its biological properties have not yet been elucidated. I believe that this should be highlighted in order to encourage the scientific community to investigate this compound with regard to its biological properties. Please see here the first report on C-glucosyl resveratrol and consider citing it in your discussion: https://www.nature.com/articles/s41598-021-83032-3

Response 1: Thank you for pointing out the critical issue. The correction done in the sentences, references added and description done in the conclusion section (lines 37-39) AND (937-944).

Comment 2: Page 1, line 42: “disease” should be replaced with “diseases” and “women” with “women’s”.

Response 2: Thank you for the comment. Comment done (line 52).

Comment 3: Figure 1. “Glucosydase” or, correctly spelled “Glucosidase” is an enzyme that cleaves a glycosidic bond, it does not catalyze glucosylation reactions. Also, the glucosyl moiety is not correctly drawn: it is missing the endocyclic oxygen, and the C-OH bonds have to be parallel to those in the 6-membered ring (i.e., C-2 is correct; C-3, C-4 and C-6 are incorrect). Please correct accordingly.

Response 3: Corrections done in the light of your comment and revised figure replaced.

Comment 4: Page 2, line 46: Please replace “are provided” with “are disclosed” or an equivalent expression.

Response 4: Thank you. Done (line 56).

Comment 5: Page 2, lines 55-56: This sentence is redundant, please consider removing it.

Response 5: The sentence is removed and a new description added.

Comment 6: Page 3, line 88: Please replace “comprised” with “orchestrated” or an equivalent expression.

Response 6: Done (line 100).

Comment 7: Page 4, line 166 and following lines: From here, the authors are describing the apoptotic effects of PD in cancer cell models, which makes its inclusion in the “Anti-apoptotic effects” section inappropriate. Please include this paragraph in the “Anticancer effects” section.

Response 7: The paragraph is now included in “Anticancer effects” section (lines 231-234).

Comment 8: Page 5, line 178: As per my previous comment, this sentence does not make sense. PD seems to exert anti-apoptotic activity in damaged tissues, but pro-apoptotic activity in cancer cells. I recommend revising this summary of the section.

Response 8: Revision done in the light of your comment.

Comment 9: Page 5, line 185-187. This is a literal repetition of page 1, line 40. Please rephrase and/or abbreviate

Response 9: Rephrased now.

Comment 10: Page 5, line 198: Reference 78 (Higuera-Ciapara, I.; Felix-Valenzuela, L.; Goycoolea, F., Astaxanthin: a review of its chemistry and applications. Critical reviews 1124 in food science and nutrition 2006, 46, (2), 185-196) revises the bioactivity of astaxanthin, not polydatin.

Response 10: This reference deleted.

Comment 11: Page 5, line 206: “...as well as induction of apoptosis…” this should be connected to the pro-apoptotic effects of PD in cancer cell lines, according to my suggestion in comment 7.

Response 11: The non-related sentences deleted.

Comment 12: Page 5, line 214: in “2-deoxy-D-glucose”, “D” should be written in small capitals.

Response 12: Done.

Comment 13: Table 1 and following tables: “increase” and “deacrease” needs to be further specified for the meaning of the arrows. Is it increased/decreased protein expression? Or increased/decreased activity?

Response 13: The meaning of arrows is more specified by adding expression, activity, level, etc.

Comment 14: Page 12, line 422: Reference 136 (Zhang, X.; Liu, X. D.; Xian, Y. F.; Zhang, F.; Huang, P. Y.; Tang, Y.; Yuan, Q. J.; Lin, Z. X., Berberine enhances survival and axonal 1259 regeneration of motoneurons following spinal root avulsion and re-implantation in rats. Free radical biology & medicine 2019, 143, 1260 454-470) focuses on berberin, not PD.

Response 14: Non-related reference deleted.

Comment 15: Page 21, line 670: “rheumatoid disease” should be replaced with “rheumatoid diseases”, plural.

Response 15: Correction done.

Comment 16: Page 23, line 731: “women health” should be replaced with “women’s health”.

Response 16: Thank you. Correction done (line 743).

Comment 17: Figure 1: Two yellow PD icons side by side (affecting Sirt1 and ERK) are not needed. Please delete one of them and have both arrows coming from the same icon. Also, please see comment 18.

Response 17: Figure corrected.

Comment 18: Final general comment: Something of extreme importance that is not discussed in the article is the fact that resveratrol is a PAIN (Pan-Assay Interference Compound) that works as a membrane disruptor by reducing the membrane dipole potential, which may affect the activation/inactivation of intracellular signalling pathways in a non-specific way, when used in cell-based assays. Thus, most results presented in this review have to be safeguarded with the fact that PD presents huge concerns when regarded as a lead for pharmaceutical development. O-glycosides of known PAINS are not likely to behave differently, as shown by Rauter et al. (https://doi.org/10.1038/s41598-021-83032-3). This has to be clear throughout, or in a dedicated section, as well as in Figure 4. C-glucosyl resveratrol, on the other hand, no longer presents this PAINS issue. Again, I strongly advise the authors to develop this topic in their discussion.

Response 18: We appreciate the critical comment. The PAIN is now added in the conclusion.

Round 2

Reviewer 1 Report

"Comment 4: Conclusions Should state the need of promoting Polydatin in industrial, commercial and research sectors.

Response 4: Thank you. It is stated in conclusion (lines 923-925).

1. Negrea B-M, Stoilov-Linu V, Pop C-E, Deák G, Crăciun N, FăgăraÈ™ MM. Expansion of the Invasive Plant Species Reynoutria japonica Houtt in the Upper BistriÈ›a Mountain River Basin with a Calculus on the Productive Potential of a Mountain Meadow. Sustainability. 2022; 14(9):5737. https://doi.org/10.3390/su14095737

2. Liu, C., Wang, W., Zhang, K., Liu, Q., Ma, T., Tan, L., & Ma, L. (2022). Protective Effects of Polydatin from Grapes and Reynoutria japonica Houtt. on Damaged Macrophages Treated with Acetaminophen. Nutrients, 14(10), 2077. https://doi.org/10.3390/nu14102077

3. Liu, H., Zhao, S., Zhang, Y., Wu, J., Peng, H., Fan, J., & Liao, J. (2011). Reactive oxygen speciesmediated endoplasmic reticulum stress and mitochondrial dysfunction contribute to polydatininduced apoptosis in human nasopharyngeal carcinoma CNE cells. Journal of Cellular Biochemistry, 112."

Comment 4 is not made properly, the statements don't match the actions.

I don't see reference [1] being mentioned in the manuscript, add it at line 30 " Reynoutria japonica is an invasive plant from the Far East.." and/or at conclusions to state its potential for polydatin sourcing in pharma industry, being an useful highly invasive plant.

Effect of polydatin on damaged macrophages treated with Acetaminophen [2] has been insufficiently described, please properly describe the mechanism involved.

1. Negrea B-M, Stoilov-Linu V, Pop C-E, Deák G, Crăciun N, FăgăraÈ™ MM. Expansion of the Invasive Plant Species Reynoutria japonica Houtt in the Upper BistriÈ›a Mountain River Basin with a Calculus on the Productive Potential of a Mountain Meadow. Sustainability. 2022; 14(9):5737. https://doi.org/10.3390/su14095737

2. Liu, C., Wang, W., Zhang, K., Liu, Q., Ma, T., Tan, L., & Ma, L. (2022). Protective Effects of Polydatin from Grapes and Reynoutria japonica Houtt. on Damaged Macrophages Treated with Acetaminophen. Nutrients, 14(10), 2077. https://doi.org/10.3390/nu14102077

Author Response

Reviewer #1

Comment 1: Comment 4 is not made properly, the statements don't match the actions.

I don't see reference [1] being mentioned in the manuscript, add it at line 30 " Reynoutria japonica is an invasive plant from the Far East.." and/or at conclusions to state its potential for polydatin sourcing in pharma industry, being an useful highly invasive plant.

Effect of polydatin on damaged macrophages treated with Acetaminophen [2] has been insufficiently described, please properly describe the mechanism involved.

  1. Negrea B-M, Stoilov-Linu V, Pop C-E, Deák G, Crăciun N, FăgăraÈ™ MM. Expansion of the Invasive Plant Species Reynoutria japonica Houtt in the Upper BistriÈ›a Mountain River Basin with a Calculus on the Productive Potential of a Mountain Meadow. Sustainability. 2022; 14(9):5737. https://doi.org/10.3390/su14095737
  2. Liu, C., Wang, W., Zhang, K., Liu, Q., Ma, T., Tan, L., & Ma, L. (2022). Protective Effects of Polydatin from Grapes and Reynoutria japonica Houtt. on Damaged Macrophages Treated with Acetaminophen. Nutrients, 14(10), 2077. https://doi.org/10.3390/nu14102077

Response 1: Thank you for the comment. The reference (Negrea B-M et al) is now mentioned twice in the manuscript, introduction (line 32) and conclusion (line 927). The reference (Liu, C. et al) is now completely described with the involved mechanisms (lines 453-456).

 Regards 

Reviewer 3 Report

Dear authors,

Thank you for your corrections. Please take a careful look at some additional comments and take your time with the corrections to avoid another round of revision:

  1. My previous comment 1 was not properly addressed. The sentence “possesses better bioavailability, and easily absorbed from the intestinal tract due to the more solubility and thereby regulates such rapid metabolism occurred in resveratrol” is incorrect. As per my previous comment “Polyphenol O-glycosides are more soluble in water than their aglycones, which is advantageous for compound dissolution and overall bioavailability, but they are actually hydrolyzed in the gut. It is mostly the aglycone that passes through the intestinal barrier into the bloodstream.“ What I am saying here is that even though glycosylation helps in dissolution, the compound will mostly reach the bloodstream as the aglycone, because the sugar is cleaved in the gut. This is not reflected in your new sentence. Also, “the more solubility” is not correct; the right expression is “its higher solubility”. The authors should furthermore make clear that this comparison is made with resveratrol (i.e. the aglycone).

  1. In light of my previous comment 3, maybe I was not completely clear. All C-OH bonds were corrected, but the same correction is needed for all atoms linked to the central 6-membered ring of the sugar. Please redraw the bond between C-5 and C-6 accordingly.

  1. Page 5, line 179: I still cannot see how the described effects are ambivalent. These effects are not contradictory, they seem to support each other, in my opinion. Please consider revising this sentence once more. Plus “... effect on apoptotic activity” sounds off. I would consider something along the lines of “Taken together, the described anti-apoptotic activities of polydatin indicate that this compound could be considered as a promising agent for the rescue of apoptosis-induced tissue damage.”

  1. Regarding my previous comment 10, the authors cannot simply remove the non-related reference, they have to provide evidence for the sentence “From the mechanistic point, polydatin modulates oxidative stress to decrease carcinogenesis and mutagenesis.”

  1. Regarding my previous comment 14, the authors cannot simply remove the non-related reference, they have to provide evidence for the sentence “Similarly, polydatin inhibited NF-κB signaling pathway contributing to the suppression of inflammatory responses and the development of pulmonary fibrosis induced by Mycoplasma pneumoniae infection. Inhibition of NLRP3 inflammasome by polydatin is the other mechanism that is employed by polydatin against Mycoplasma pneumoniae infection.”

  1. Page 30, line 937: Please note that PAINS is not a phenomenon, and thus should not be referred to as “the PAINS”. PAINS are a class of compounds that raise concerns from the 'target selectivity' point of view. Some critical issues to be addressed:

    1. Today the Pan-Assay Interference Compounds (PAINS) has been proposed as a threat to recognition bioactivity of natural compounds, including compounds such as resveratrol.“ This sentence needs a reference. Also “recognition bioactivity” is hard to understand. Please make your idea clearer. 

    2. PAINS cause disruption in the membrane by reducing the bipolar potential of the membrane and as a result can affect intracellular signaling pathways.” Please read the appropriate references before discussing this issue (you can find them cited here: DOI: 10.1038/513481a). There are several types of PAINS. Resveratrol is a membrane disruptor and, as it is, the authors’ sentence is misleading. Also, “bipolar potential of the membrane “is incorrect, it should read “membrane dipole potential”. Finally, it is important to add a disclosure at this point stating that since PAINS-type behavior “...can affect intracellular signaling pathways” in a non-specific way, the cell-based data presented in this review might be a product of such effects and not caused by specific binding to therapeutic targets. Thus, cell-based data should be interpreted with caution.

    3. Recent studies have shown that C-glucosylation of polyphenol compounds causes the PAINS effect of this group of compounds to disappear, while the O-glucosylation has no effect on this behavior [242, 243].” The term “disappear” is not the most appropriate technical term, please rephrase. Also, reference 243 is not related to the study showing the effects of C-glucosylation on PAINS-type behavior. This reference deals with the differences between O- and C-glucosyl flavone metabolism, it should be cited only if the authors wish to reinforce the benefits of inserting a C-glucosyl moiety to resveratrol instead of the O-glucosyl one in PD in addition to helping with the PAINS-related issue (which I strongly encourage).

    4. Polydatin, an O-glucosylated derivative of resveratrol, has the same concerns, and more research should be done to improve associated PAINS behavior.” A more careful approach to this idea is needed here. Please rephrase to something along the lines of “As per previous data [242], polydatin, an O-glucosylated derivative of resveratrol, is expected to present the same concerns as the aglycone with regards to inducing non-specific reductions in the membrane dipole potential. Therefore, in order to assess if C-glucosylation of resveratrol is able to preserve the promising bioactivities of polydatin described in this review while addressing the mentioned PAINS-related issue, the biological evaluation of this C-glucosyl analog of polydatin is of utmost importance.”

    5. C-glucosylation and O-glucosylation should have the C- and O- letters in italic.

Kind regards.

Author Response

Reviewer #3

Overall comment: Dear authors,

Thank you for your corrections. Please take a careful look at some additional comments and take your time with the corrections to avoid another round of revision:

Response: Thank you again for the comments. The “point-by-point” and itemized modification has been done in the light of your comments.

Comment 1: My previous comment 1 was not properly addressed. The sentence “possesses better bioavailability, and easily absorbed from the intestinal tract due to the more solubility and thereby regulates such rapid metabolism occurred in resveratrol” is incorrect. As per my previous comment “Polyphenol O-glycosides are more soluble in water than their aglycones, which is advantageous for compound dissolution and overall bioavailability, but they are actually hydrolyzed in the gut. It is mostly the aglycone that passes through the intestinal barrier into the bloodstream.“ What I am saying here is that even though glycosylation helps in dissolution, the compound will mostly reach the bloodstream as the aglycone, because the sugar is cleaved in the gut. This is not reflected in your new sentence. Also, “the more solubility” is not correct; the right expression is “its higher solubility”. The authors should furthermore make clear that this comparison is made with resveratrol (i.e. the aglycone).

Response 1: The previously non-correct sentence is now deleted.

Comment 2: In light of my previous comment 3, maybe I was not completely clear. All C-OH bonds were corrected, but the same correction is needed for all atoms linked to the central 6-membered ring of the sugar. Please redraw the bond between C-5 and C-6 accordingly.

Response 2: Thank you for the precise comment. Correction done between C-5 and C-6.

Comment 3: Page 5, line 179: I still cannot see how the described effects are ambivalent. These effects are not contradictory, they seem to support each other, in my opinion. Please consider revising this sentence once more. Plus “... effect on apoptotic activity” sounds off. I would consider something along the lines of “Taken together, the described anti-apoptotic activities of polydatin indicate that this compound could be considered as a promising agent for the rescue of apoptosis-induced tissue damage.”

Response 3: We did our best to revise the aforementioned points, and cited our previously published articles to describe the interconnections between pathways (lines 178-187).

Comment 4: Regarding my previous comment 10, the authors cannot simply remove the non-related reference, they have to provide evidence for the sentence “From the mechanistic point, polydatin modulates oxidative stress to decrease carcinogenesis and mutagenesis.”

Response 4: We previously removed the non-related reference “Higuera-Ciapara, I.; Felix-Valenzuela, L.; Goycoolea, F., Astaxanthin: a review of its chemistry and applications. Critical reviews 1124 in food science and nutrition 2006, 46, (2), 185-196”. In the second version of revision we added related reference. However, additional references are describing these effects in the next lines.

Comment 5: Regarding my previous comment 14, the authors cannot simply remove the non-related reference, they have to provide evidence for the sentence “Similarly, polydatin inhibited NF-κB signaling pathway contributing to the suppression of inflammatory responses and the development of pulmonary fibrosis induced by Mycoplasma pneumoniae infection. Inhibition of NLRP3 inflammasome by polydatin is the other mechanism that is employed by polydatin against Mycoplasma pneumoniae infection.”

Response 5: We previously removed the non-related reference “(Zhang, X.; Liu, X. D.; Xian, Y. F.; Zhang, F.; Huang, P. Y.; Tang, Y.; Yuan, Q. J.; Lin, Z. X., Berberine enhances survival and axonal 1259 regeneration of motoneurons following spinal root avulsion and re-implantation in rats. Free radical biology & medicine 2019, 143, 1260 454-470)”. In the second version of revision we added related reference.

Comment 6: Page 30, line 937: Please note that PAINS is not a phenomenon, and thus should not be referred to as “the PAINS”. PAINS are a class of compounds that raise concerns from the 'target selectivity' point of view. Some critical issues to be addressed:

  1. “Today the Pan-Assay Interference Compounds (PAINS) has been proposed as a threat to recognition bioactivity of natural compounds, including compounds such as resveratrol.“ This sentence needs a reference. Also “recognition bioactivity” is hard to understand. Please make your idea clearer.

Reference: Reference added and correction done.

  1. “PAINS cause disruption in the membrane by reducing the bipolar potential of the membrane and as a result can affect intracellular signaling pathways.” Please read the appropriate references before discussing this issue (you can find them cited here: DOI: 10.1038/513481a). There are several types of PAINS. Resveratrol is a membrane disruptor and, as it is, the authors’ sentence is misleading. Also, “bipolar potential of the membrane “is incorrect, it should read “membrane dipole potential”. Finally, it is important to add a disclosure at this point stating that since PAINS-type behavior “...can affect intracellular signaling pathways” in a non-specific way, the cell-based data presented in this review might be a product of such effects and not caused by specific binding to therapeutic targets. Thus, cell-based data should be interpreted with caution.

Reference: Correction and disclosure done (lines 947-952).

  1. “Recent studies have shown that C-glucosylation of polyphenol compounds causes the PAINS effect of this group of compounds to disappear, while the O-glucosylation has no effect on this behavior [242, 243].” The term “disappear” is not the most appropriate technical term, please rephrase. Also, reference 243 is not related to the study showing the effects of C-glucosylation on PAINS-type behavior. This reference deals with the differences between O- and C-glucosyl flavone metabolism, it should be cited only if the authors wish to reinforce the benefits of inserting a C-glucosyl moiety to resveratrol instead of the O-glucosyl one in PD in addition to helping with the PAINS-related issue (which I strongly encourage).

Response: Reference 243 (newly 251) deleted.

  1. “Polydatin, an O-glucosylated derivative of resveratrol, has the same concerns, and more research should be done to improve associated PAINS behavior.” A more careful approach to this idea is needed here. Please rephrase to something along the lines of “As per previous data [242], polydatin, an O-glucosylated derivative of resveratrol, is expected to present the same concerns as the aglycone with regards to inducing non-specific reductions in the membrane dipole potential. Therefore, in order to assess if C-glucosylation of resveratrol is able to preserve the promising bioactivities of polydatin described in this review while addressing the mentioned PAINS-related issue, the biological evaluation of this C-glucosyl analog of polydatin is of utmost importance.”

Response: Done (lines 953-958).

  1. C-glucosylation and O-glucosylation should have the C- and O- letters in italic.

Response: Thank you. Done.

Regards